# Social memory deficit caused by dysregulation of the cerebellar vermis

Owen Y. Chao[1], Salil Saurav Pathak[1,8], Hao Zhang [1,8], George J. Augustine [2], Jason M. Christie[3], Chikako Kikuchi[4], Hiroki Taniguchi[5,6] & Yi-Mei Yang [1,7] ✉

Social recognition memory (SRM) is a key determinant of social interactions. While the cerebellum emerges as an important region for social behavior, how cerebellar activity affects social functions remains unclear. We selectively increased the excitability of molecular layer interneurons (MLIs) to suppress Purkinje cell firing in the mouse cerebellar vermis. Chemogenetic perturbation of MLIs impaired SRM without affecting sociability, anxiety levels, motor coordination or object recognition. Optogenetic interference of MLIs during distinct phases of a social recognition test revealed the cerebellar engagement in the retrieval, but not encoding, of social information. c-Fos mapping after the social recognition test showed that cerebellar manipulation decreased brain-wide interregional correlations and altered network structure from medial prefrontal cortex and hippocampus-centered to amygdala-centered modules. Anatomical tracing demonstrated hierarchical projections from the central cerebellum to the social brain network integrating amygdalar connections. Our findings suggest that the cerebellum organizes the neural matrix necessary for SRM.

Social behavior, defined as interactions among conspecifics, is evolutionarily conserved and critical for survival of both individuals and species. Social behavior involves processes that detect, store, and respond to social stimuli, which requires concerted actions of multiple brain areas including the medial prefrontal cortex (mPFC), anterior cingulate cortex (ACC), hippocampus, and amygdala[1–3]. Damage to any of these regions can affect social behavior, making it susceptible to pathological conditions, e.g., Alzheimer's disease, schizophrenia, and autism spectrum disorder (ASD)[4–6]. Despite its significance, our understanding of the neural basis for social behavior is incomplete.

The cerebellum, which is not typically considered to be part of the social brain[1–3], has emerged as a key node in an integrative network that links diverse functions of sense, motor, emotion, cognition and working memory[7–9]. In the cerebellar cortex, Purkinje cells (PCs) receive excitatory and inhibitory inputs and carry the sole output.

While synaptic excitation facilitates PC firing, the PC firing activity is strongly controlled by feedforward inhibition from molecular layer interneurons (MLIs)[10]. Nearly all PCs send GABAergic axons to inhibit the cerebellar nuclei (CN)[11]. Neurons in the CN directly project to subcortical regions (first-order targets) such as the ventral thalamus (vTH). Via the first-order neurons, the CN connects to second-order targets virtually in the entire neocortex[12]. The neocortex, in turn, sends input back to the cerebellar cortex via the pontine nuclei. Through these feedback loops, the cerebellum is viewed as an estimator of the internal state for the operation of movement and cognition[13]. The diverse functionality of the cerebellum is thought to come from the topographical organization of the cerebello-cortical circuits. Functional magnetic resonance imaging (fMRI) of the human brain has revealed that lobules I-V and VIII are involved in sensorimotor tasks via interactions with the somatosensory and motor cortices; lobules VI-VII

[1]Department of Biomedical Sciences, University of Minnesota Medical School, Duluth, MN 55812, USA. [2]Lee Kong Chian School of Medicine, Nanyang Technological University, 308232 Singapore, Singapore. [3]University of Colorado School of Medicine, Aurora, CO 80045, USA. [4]Max Planck Florida Institute for Neuroscience, Jupiter, FL 33458, USA. [5]Department of Pathology, Ohio State University Wexner Medical Center, Columbus, OH 43210, USA. [6]Chronic Brain Injury, Ohio State University Wexner Medical Center, Columbus, OH 43210, USA. [7]Department of Neuroscience, University of Minnesota, Minneapolis, MN 55455, USA. [8]These authors contributed equally: Salil Saurav Pathak, Hao Zhang. ✉e-mail: ymyang@d.umn.edu

in social cognition, working memory, and language via interactions with the prefrontal, temporal and parietal cortices; lobules IX and X (flocculus) in eye movement and balance via interactions with the vestibular system[14,15]. However, these functions are generalized because motor and non-motor dichotomy of certain lobules can occur[16]. The anterior and posterior lobes are further divided into a central vermis and lateral hemispheres. The vermis and its projections to the fastigial nuclei (FN) in the medial CN receive input from the spinocerebellar tract and send output to the brainstem, thalamus, and cerebral cortex. In addition to motor coordination and postural control, a characteristic function of the vermis is to regulate emotion via interactions with the limbic circuits[17]. A pioneer study demonstrated that stimulating the vermis, but not the hemispheres, generated responses in the limbic regions in animals[18]. Work with neuronal tracing further shows the FN directly projects to the thalamus and hypothalamus[19–22], while its links to the hippocampus and amygdala may be indirect[21]. From the thalamus, the FN connects to the limbic cortex including the mPFC and ACC[21,22]. This provides a structural base for the vermis-dependent aggression, spatial memory, and social association[23–25].

Social interactions rely on intentions, emotions, and memories of social encounters, covering distinct yet interactive processes of social cognition and social memory[1–3]. Social cognition is the ability to imitate the actions of others (mirroring) and to understand the mental states of oneself and others (mentalizing)[1–3]. Social memory (or social recognition memory, SRM) is the ability to distinguish familiar from novel conspecifics by recalling previous encounters; it is supported by the integrity of the mPFC, ACC, hippocampus, and amygdala[26–30]. fMRI studies suggest the human cerebellum partakes in the neural networks for social cognition[31,32]. While there is no direct evidence for its involvement in SRM, the cerebellum is known to encode procedural memories, most notably eyeblink conditioning[33]. Early reports indicate MLIs contribute to the consolidation of learned eyeblink response by suppressing PC activity and releasing the CN from tonic PC inhibition to enable the eyeblink reflex[34–37]. Blocking MLI-PC synaptic transmission or excitation of MLIs in the floccular lobe alters the adaptation of the vestibulo-ocular reflex[38,39]. Inhibiting MLIs in lobule VI affects reverse learning and in lobule VII affects novelty-seeking behavior in juvenile mice, signifying a role of MLIs in cognitive and social development beyond the motor domain[37]. Interestingly, postmortem analysis of ASD patients shows an increase of GAD67 mRNA in MLIs, implying dysregulated feedforward inhibition to PCs[40]. This is confirmed in various ASD mouse models, where low PC activity, at least partially caused by MLI over-inhibition, leads to social impairments[41–44]. However, it remains to be determined how the cerebellum transforms such abnormal local activity into social behavior deficits.

Using chemogenetic and optogenetic techniques, we mimicked the dysregulated MLI-PC inhibition observed in ASD patients[40] and mouse models[42–44] by selectively increasing the excitability of MLIs to reduce PC firing in the vermis. Targeting MLIs also helped minimize potential changes in movement[25] or aggression[23] that could confound the analysis of social interactions[45], if PCs were directly silenced. We evaluated the effects of this manipulation on mouse behaviors. As it specifically affected SRM, we conducted functional mapping of the cerebello-cortical axis by measuring the expression of an immediate early gene c-Fos after the SRM test[46], and traced the inter-nucleus connections from the FN to the cerebral cortex. Our findings indicate that MLIs fine-tune the cerebellar intrinsic circuits to contextualize the extrinsic networks for SRM, thereby enabling normal social behavior.

## Results
### Chemogenetic excitation of MLIs in the cerebellar vermis disrupts SRM
We used an adeno-associated virus (AAV)-mediated chemogenetic approach to express an excitatory Designer Receptors Exclusively Activated by Designer Drugs (DREADD, AAV8-hSyn-DIO-hM3Dq-mCherry) in major vermal lobules of the anterior (vermis IV/V) or posterior (vermis VI/VII) cerebellum of adult c-kit[IRES-Cre] mice (Fig. 1a). Via Cre-LoxP recombination, the c-kit promoter allows selective transduction of MLIs postnatally[47]. The vermis was targeted because of its association with the limbic system and neuropsychiatric disorders[17]. To examine transduction efficiency, we imaged cerebellar sections several weeks after viral infusions and behavioral tests. Despite individual variability, robust yet localized expression of the hM3Dq receptor (marked with mCherry) was observed in lobule IV/V or VI/VII (Fig. 1b, Supplementary Fig. 1). Within the individual cerebellum, the hM3Dq was mostly present in midsagittal sections and diminished in sections >500 μm away from the midline (Supplementary Fig. 1). By normalizing the area containing mCherry to the total area of lobule IV/V or VI/VII in the midsagittal sections, we found $29.2 \pm 2.7\%$ of lobule IV/V and $35.8 \pm 3.0\%$ of lobule VI/VII ($n = 6$ mice/group) were affected. Higher-resolution imaging revealed that hM3Dq expression was predominantly in MLIs (Fig. 1b) although limited Cre activity could appear in Golgi and glial cells[47]. Among MLIs, stellate cells (SCs) innervate the distal dendrites of PCs and do not directly affect PC firing, whereas basket cells (BCs) execute powerful inhibition of PC activity by forming GABAergic synapses on the soma and ephaptic connections on the axon initial segment of PCs[10]. Thus, we focused on the BC-PC microcircuits by co-labeling PCs with an anti-Calbindin antibody. As shown before[10], a BC innervated several adjacent PC somas (Supplementary Fig. 1). We validated our chemogenetic approach with ex vivo patch-clamp recordings. Activation of the hM3Dq with clozapine-N-oxide (CNO; 10 μM) significantly increased the frequency of action potentials (APs) in BCs expressing the excitatory DREADD but had no effect on non-transduced BCs (Fig. 1c, d; statistical details in Supplementary Table 1). As a result of synaptic inhibition[10,43], CNO reduced the firing rate of PCs that were connected to hM3Dq-expressing BCs and increased the coefficient of variation (CV) of inter-AP intervals in these PCs. CNO did not affect PCs innervated by non-transduced BCs (Fig. 1e, f).

Next, we evaluated the impact of the chemogenetic excitation of MLIs on mouse behavior by intraperitoneal (i.p.) injection of CNO (1 mg/kg) 30–40 min before each assay[25,43]. Compared to their littermate controls that lacked Cre recombinase, c-kit[IRES-Cre] mice transduced with the hM3Dq in lobule IV/V or VI/VII showed no differences in open-arm avoidance (measured with the elevated plus maze test), locomotion (measured with the open field test), motor coordination (measured with the rotarod test), or social play (measured with the reciprocal social interaction test) (Supplementary Fig. 2), in line with early reports that interfering vermal activity in adult mice did not affect their general performance[25,37,43]. To further scrutinize their social behavior, we conducted the three-chamber social test that consisted of a sociability trial immediately followed by a social novelty trial (Fig. 1g). In the sociability trial, all groups explored the stranger mouse more than the empty cup and rendered positive sociability indices (Fig. 1g, top). In the social novelty trial, both control mice and those expressing the DREADD in vermis IV/V explored the new stranger more than the old one, while mice expressing the DREADD in vermis VI/VII did not show such a preference, resulting in a social novelty index near 0 (Fig. 1g, bottom).

Social novelty preference relies on remembering previous encounters and is driven by seeking the unknown[48]. To determine whether the lack of social novelty preference was caused by memory loss, we performed the social recognition test with a prolonged interval (45 min) between a learning and a testing trial (Fig. 1h). The time delay was introduced to assess memory retention. In the learning trial, all animals displayed normal social tendency toward the stranger mouse and had positive social learning scores (Fig. 1h, top), consistent with findings from the three-chamber social test (Fig. 1g). In the testing trial, the control group again showed intact social novelty preference.

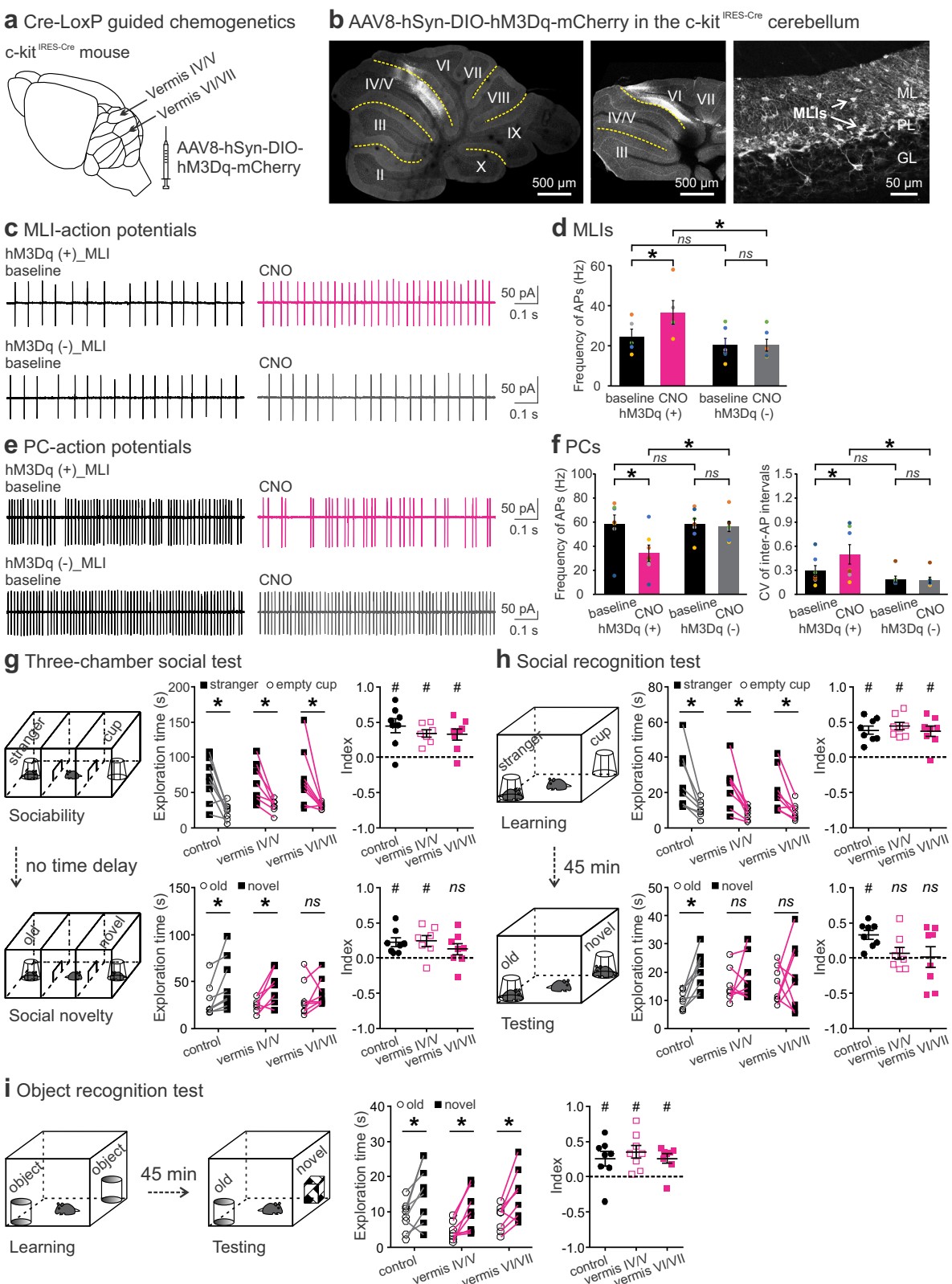

However, the vermis-perturbed groups explored the new and the old strangers indiscriminately, accompanied with low social recognition scores (Fig. 1h, bottom). To determine whether the chemogenetic perturbation reduced memory capacity or motivation to seek novelty in general, we did the object recognition test, which was identical to the social recognition test except that the social stimuli were replaced with inanimate objects (Fig. 1i). Surprisingly, all groups spent more

time exploring the novel object than the old object with positive object recognition scores (Fig. 1i), indicating well-preserved object-based memory and novelty-seeking behavior. The total exploration time among groups was comparable in all tests (Supplementary Table 1). Together, these results suggest that the vermis is important for SRM without affecting anxiety levels, locomotion, exploratory motivation, social approach, or object recognition memory in animals.

**Fig. 1 | Chemogenetic excitation of MLIs in cerebellar lobules IV-VII disrupted social, but not object, recognition memory. a** Design to target MLIs by infusing AAV8-hSyn-DIO-hM3Dq-mCherry (hM3Dq) into the anterior (lobule IV/V) or posterior (lobule VI/VII) vermis of a c-kit[IRES-Cre] mouse. **b** Examples of AAV-mediated expression of hM3Dq in MLIs (right) in lobule IV/V (left) or VI/VII (middle). ML, molecular layer; PL, Purkinje layer; GL, granular layer. Cell-attached patch-clamp recordings of APs from MLIs (**c**) expressing hM3Dq (top) or not (bottom) and their downstream PCs (**e**) before and after CNO application (10 μM). Effects of CNO on firing frequency of hM3Dq-containing ($n = 5$) or -lacking ($n = 6$) MLIs (**d**) and on firing frequency and coefficient of variation (CV) of inter-AP intervals of PCs ($n = 7$, both groups; **f**). **g** A three-chamber social test included a sociability and a social novelty trial without time delay in between (left). All groups showed intact sociability (explored the stranger more than the cup) and intact social novelty (explored the novel stranger more than the old one), except for vermis VI/VII group (right). **h** A social recognition test included a learning and a testing trial with a 45 min inter-trial interval (left). All groups displayed intact sociability, but perturbation of lobule IV/V or VI/VII impaired animals' social recognition (explored the novel and the old strangers indiscriminately) (right). **i** An object recognition test included a learning and a testing trial with a 45 min inter-trial interval (left). All groups had intact object recognition (explored the novel object more than the old one) (right). CNO (1 mg/kg) was given 30–40 min before each test. In the sociability trial of three-chamber tests and in the learning trial of social recognition tests: index = [time for exploring the stranger − time for exploring the cup] / total exploration time. In the social novelty trial of three-chamber tests and in the testing trials of social and object recognition tests: index = [time for exploring the novel one − time for exploring the old one] / total exploration time. *$p < 0.05$, two-tailed paired or unpaired $t$-test. #$p < 0.05$, two-tailed one-sample $t$-test compared to 0. ns, not significant. $n = 7$–9 mice/group for behavior tests. Data are presented as the mean ± SEM and the center of error bars is the mean. Source data are provided as a Source Data file.

## Optogenetic stimulation of MLIs in the vermis interferes with retrieval, but not encoding, of SRM

SRM is a type of declarative memory that consists of encoding, storage/consolidation, and retrieval of social information[49]. To specify at which of these stages the cerebellum is engaged, we employed an optogenetic approach that permits precise temporal control of neuronal firing with a bacterial artificial chromosome (BAC) transgenic mouse line that uses the *nNOS* promoter to express channelrhodopsin-2 (ChR2) in MLIs (nNOS-ChR2)[50]. ChR2 (fused to YFP) was found in most MLIs in the cerebellum (Fig. 2a). Co-labeling PCs with an anti-Calbindin antibody confirmed that ChR2 was exclusively present in the somas, dendrites, and axon terminals of MLIs. To measure neuronal responses to photostimulation, we recorded APs from MLIs (BCs) and PCs while delivering an 8 Hz train of light pulses (470 nm, 25 ms duration) (Fig. 2b). This low frequency ensures reliable generation of APs in response to light flashes;[50] and it falls within the range of theta oscillations (4–10 Hz), which play a role in learning and memory[51]. Specifically, theta-burst stimulation of the human cerebellum affects episodic memory[52]. As shown in Fig. 2b–d, the photostimulation rapidly increased AP frequency in MLIs, while reducing it in postsynaptic PCs. Upon cessation, MLI and PC activity returned to baseline levels. Additionally, the photostimulation decreased regularity of AP firing in PCs, indicated by a dramatic increase in the CV of inter-AP intervals (Fig. 2e). When all inter-AP intervals were plotted on a histogram and fit with a Gaussian function, we found the photostimulation created two distinct peaks centering around 14.5 ms and 54.2 ms, in contrast to the unimodal pattern with a peak around 17.5 ms without stimulation (Fig. 2f). Based on the mean values of the Gaussian distribution, the 1st peak likely represented spontaneous PC activity whereas the 2nd peak correlated to the stimulus pattern.

Applying the same light pulses through optic fibers implanted in lobule IV/V or VI/VII, we studied the behavior consequences of photostimulating MLIs. In the open field test, nNOS-ChR2 mice and their littermate controls (not expressing ChR2) traveled a similar distance, and the photostimulation did not change their locomotor activity (Supplementary Table 1). To know how the vermis was involved in the process of SRM, we photostimulated in different sessions of the social recognition test. When the light was presented during the learning trial (Fig. 2g), all mice explored the stranger more than the empty cup with positive social learning scores (Fig. 2g, top). They also explored the novel stranger more than the familiar one with positive social recognition scores (Fig. 2g, bottom). Perturbing lobule IV/V yielded less exploration in total as compared to the other groups (Supplementary Table 1), which may reflect deficits in motivation, attention and/or sensorimotor integration associated with interference of the anterior cerebellum[25]. Overall, perturbation of the cerebellum at the encoding stage did not impair sociability, social novelty preference or SRM.

In contrast, when the light was presented during the testing trial (Fig. 2h), all mice spent more time exploring the stranger than the empty cup with positive indices of sociability (Fig. 2h, top). Yet, unlike the control group that explored the novel stranger more than the old one, the vermis-perturbed groups did not prefer the novel stranger, leading to poor indices of social recognition (Fig. 2h, bottom). As there was no group difference in total exploration in this case (Supplementary Table 1), the impaired social recognition was likely accounted for by an inability to recall previous encounters rather than by confounding factors such as a lack of inquisitiveness. When the light was presented in the testing trial of the object recognition test (Fig. 2i), all animals preferred the new object to the old one with positive indices of object recognition (Fig. 2i), implying no effect of this manipulation on non-social recognition memory. These results suggest that the vermis plays a specific role in the retrieval (or consolidation), but not encoding, of social information.

## Perturbing the vermis decreases functional connectivity and disorganizes network structure of brain regions essential for SRM

To understand how the cerebellum mediates SRM, we monitored neuronal activity via a molecular marker, c-Fos, in 24 brain regions following the social recognition test on c-kit[IRES-Cre] mice and their littermate controls (Fig. 3a). As shown in Fig. 1h, activation of vermal MLIs by the excitatory DREADD disrupted SRM. 90 min after the test, we collected brain tissue for c-Fos immunostaining. Basal levels of c-Fos are normally low but can be elevated by external stimuli, making it a useful tool to identify activated neurons[46]. The 90 min window was chosen because c-Fos expression peaks 1-2 h after stimulation[53]. We examined c-Fos in brain regions with well-established roles in SRM[26–30]. By counting the number of c-Fos-positive cells in each region and normalizing these values to the mean number of c-Fos-positive cells in the control group, we found significant effects on c-Fos levels for the mPFC, parahippocampal cortices, amygdala, hypothalamus, and ventral tegmental area (VTA) (Supplementary Table 1). Perturbing the vermis, irrespective of the lobules, elevated c-Fos expression in the temporal association cortex (TeA), ectorhinal cortex (Ect), and perirhinal cortex (PRh) compared to the control group (Supplementary Fig. 3). These parahippocampal cortices interact with the hippocampus and cerebral cortex to support high-order cognitive processes such as contextual associations and emotional inferences[54]. In other regions, changes in c-Fos depended upon which lobule was targeted. For instance, perturbing lobule IV/V mostly reduced c-Fos levels in subdivisions of the mPFC (anterior cingulate cortex (rostral), Acg: $p = 0.024$; prelimbic cortex, PL: $p = 0.044$; but not infralimbic cortex, IL: $p > 0.05$; Fig. 3b). In contrast, perturbing lobule VI/VII increased c-Fos levels in subdivisions of the amygdala (central nucleus of amygdala, CeA: $p = 0.003$; basolateral amygdala, BLA: $p = 0.007$; but not basomedial amygdala, BMP: $p > 0.05$; Fig. 3c). The differences are likely related to distinct connections of the anterior and posterior cerebellum to the cortical and subcortical regions[14,15].

**a** nNOS-ChR2-YFP BAC mouse

**b** Light OFF    Light ON (8 Hz)    Light OFF

**c** MLIs   **d** PCs   **e** PCs   **f** PCs

**g** Social recognition test (light ON in learning trial)

**h** Social recognition test (light ON in testing trial)

**i** Object recognition test (light ON in testing trial)

To identify subtypes of the c-Fos-positive cells, we co-labeled glutamatergic and GABAergic neurons with anti-CaMKII and anti-GAD67 antibodies, respectively. As illustrated in Supplementary Fig. 4, c-Fos-positive cells in these regions were primarily glutamatergic neurons (~60–80%), with small proportions of GABAergic neurons (~10–20%) and other cell types (~10–20%). This result suggests that while various cells are activated, excitatory neurons that typically represent the output from each region play a major role in mediating the cellular response to SRM.

To gain insight into functional connectivity, we calculated Pearson correlation coefficients ($r$) of each pair of brain regions across the subjects in each group based on the c-Fos measurements. We then created interregional correlation matrices with the values of Pearson's $r$ from 1 to −1 indicated by a gradient spectrum from red to indigo,

**Fig. 2 | Optogenetic stimulation of MLIs in cerebellar lobules IV-VII in the retrieval, but not encoding, phase impaired social recognition memory.**
**a** Cerebellar slice from a nNOS-ChR2 mouse (left). Co-labeling of PCs with an anti-Calbindin antibody showed selective expression of ChR2 in the soma and processes of MLIs (right). ML molecular layer, PL Purkinje layer, GL granular layer, ChR2 channelrhodopsin-2, YFP yellow fluorescent protein. **b** APs recorded from MLIs (middle) and PCs (bottom) of a nNOS-ChR2 mouse in response to photostimulation (25 ms, 8 Hz, top). Photostimulation increased the AP frequency from MLIs ($n = 7$; **c**) but decreased it from PCs ($n = 9$; **d**) and increased the coefficient of variation (CV) of inter-AP intervals for PCs ($n = 9$; **e**). Histograms of the number of events versus inter-AP interval (bin width 1 ms) showed a shift from unimodal to bimodal distribution by photostimulation (**f**). Dotted lines were fits with a Gaussian function: $f(x) = a \cdot e^{-(x-\mu)^2/2\sigma^2}/(\sigma\sqrt{2\pi}) + c$. $a$, height of the peak; $\mu$, center position of the peak; $\sigma$, standard deviation. **g** In a social recognition test, light delivery (25 ms, 8 Hz, 10 s pause every 50 s) in the learning trial did not affect animals' social approach

(explored the stranger more than the cup) or social recognition (explored the novel stranger more than the old one). **h** Same light delivery in the testing trial impaired animals' social recognition (explored the novel and the old strangers equally) without affecting their social preference (explored the stranger more than the cup). **i** Same light delivery in the testing trial of an object recognition test had no effect on animals' object recognition (explored the novel object more than the old one). In the learning trial of social recognition tests: index = [time for exploring the stranger − time for exploring the cup] / total exploration time. In the testing trial of social and object recognition tests: index = [time for exploring the novel one − time for exploring the old one] / total exploration time. Positive values of indices suggest intact performance. *$p < 0.05$, two-tailed paired $t$-test. #$p < 0.05$, two-tailed one-sample $t$-test compared to 0. ns, not significant. $n = 7$–9 mice/group for behavior tests. Data are presented as the mean ± SEM and the center of error bars is the mean. Source data are provided as a Source Data file.

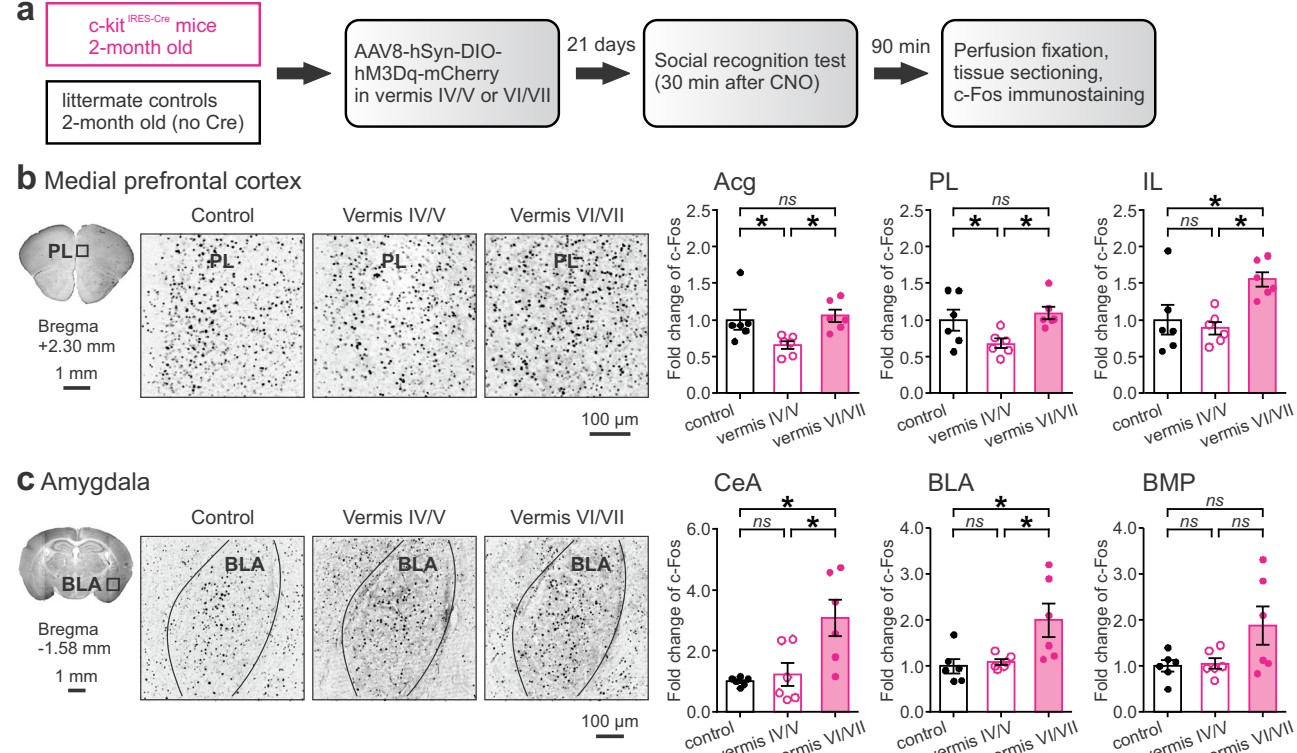

**Fig. 3 | Altered c-Fos expression by chemogenetic excitation of MLIs in cerebellar lobules IV-VII following the social recognition test. a** Design of c-Fos imaging after the social recognition test. Examples of c-Fos staining (black dots) in the PL (**b**) and BLA (**c**) subregions. Fold change, defined as the number of c-Fos-positive cells in each group divided by the average value of the control group, was summarized for all subregions in the medial prefrontal cortex and amygdala.

*$p < 0.05$, two-tailed Fisher's LSD test. ns, not significant. $n = 6$ mice/group. Data are presented as the mean ± SEM and the center of error bars is the mean. Source data are provided as a Source Data file. Acg anterior cingulate cortex (rostral), PL prelimbic cortex, IL infralimbic cortex, CeA central nucleus of amygdala, BLA basolateral amygdala, BMP basomedial amygdala.

respectively (Fig. 4a). As the matrices exhibited discrete patterns for each group, we further analyzed the network connectivity by only including the most positive ($r > 0.6$) and the most negative ($r < −0.6$) correlations in the network plots (Fig. 4b). The plots comprised the brain regions (nodes) and their connections (edges; red lines: $r > 0.6$, indigo lines: $r < −0.6$). A chi-square test revealed that the control group had more edges, and proportionally more positive correlations, than the vermis-perturbed groups (vermis IV/V: $\chi^2 = 66.03$, $p < 0.001$, vermis VI/VII: $\chi^2 = 27.42$, $p < 0.001$; Fig. 4c). Perturbing the vermis reduced the mean $r$ of all brain regions compared to the control group (Fig. 4d), implying a decrease in concerted brain-wide activity. Figure 4e–h highlighted four key regions for SRM. Both vermis groups had lower $r$ than the control group in the mPFC, ACC, and hippocampus, except that the difference for the vermis VI/VII group in

the ACC was not significant ($p = 0.097$). Conversely, the vermis groups had higher $r$ than the control group in the amygdala though the increase of the vermis IV/V group was mild ($p = 0.079$); this may reflect the lobule-specific changes of c-Fos expression in the amygdala (Fig. 3c).

To define network properties, we did graph theoretical analysis by focusing on the positive correlations in the matrix of each group. First, we computed degree (the number of edges that a node has) and betweenness (the number of shortest paths that pass through a node) to assess the centrality of a network node. A high value for "degree" indicates a brain region that is connected to many others, and a high "betweenness" indicates a brain region that has close control over other regions[55,56]. For degree, the Ect, hippocampus (CA1 and dentate gyrus [DG]), and primary motor cortex (M1) were ranked high (>0.8) in

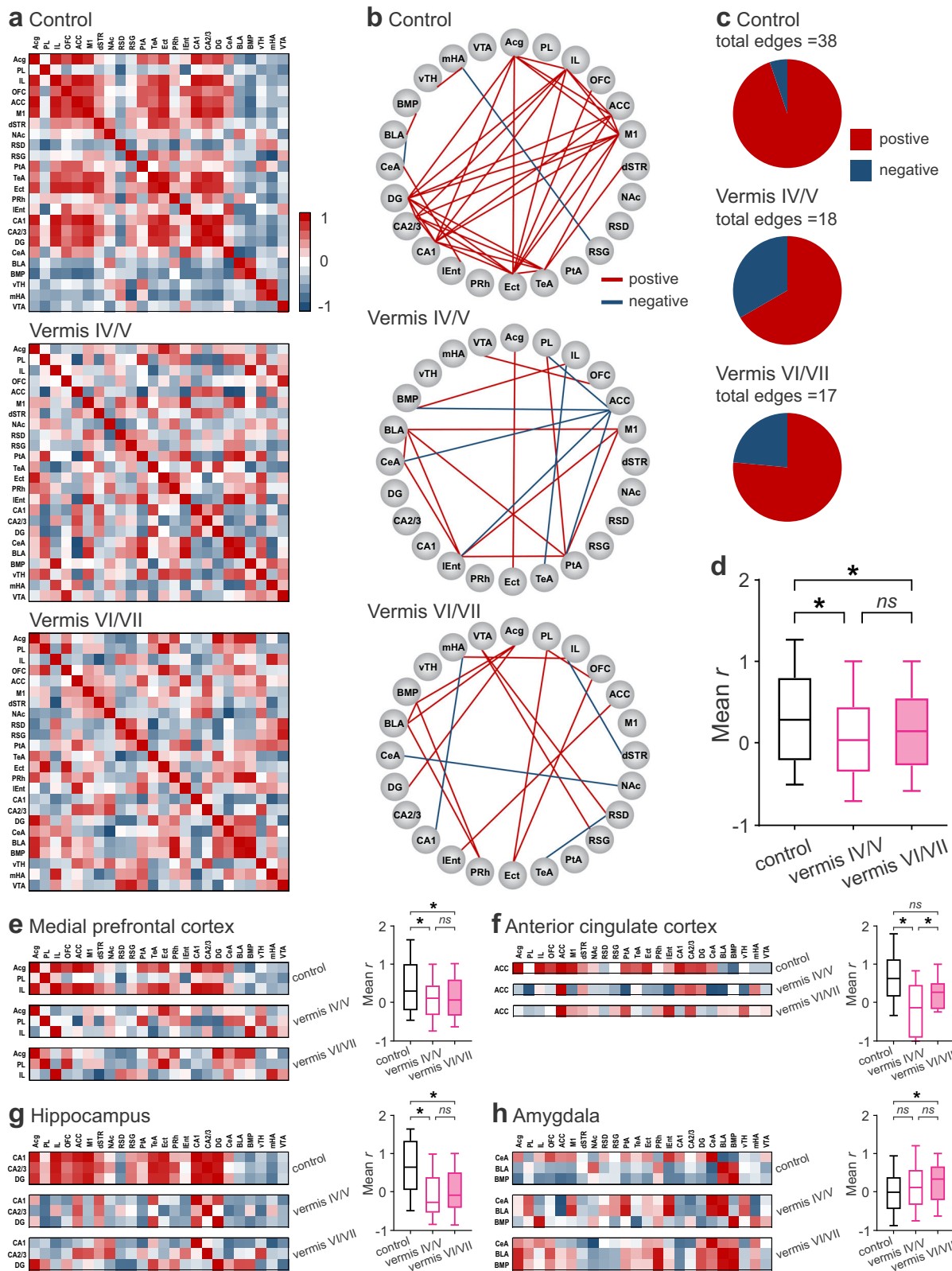

the control group, while the amygdala (BLA and BMP), parietal asso-ciation cortex (PtA), lateral entorhinal cortex (lEnt), and Acg were ranked high in the vermis groups (Fig. 5a–c, left). For betweenness, the Ect and TeA were ranked high (>0.8) in the control group, while the PtA and Acg were ranked high in the vermis groups (Fig. 5a–c, middle). Next, we quantified within-community Z-scores (the extent of a node connected in its own community) and participation coefficients

(the extent of a node connected to other communities) to outline the modularity of the network. A brain region with high rankings in both Z-score and participation coefficient is classified as a connector hub that regulates the interactions within and between communities (modules)[55,56]. In the control group, the Ect, TeA, hippocampus (CA1, CA2/3 and DG), mPFC (Acg and IL), M1, and ACC had high ranks for Z-scores and participation coefficients, whereas in the vermis groups,

**Fig. 4 | Chemogenetic excitation of MLIs in cerebellar lobules IV-VII reduced interregional connectivity activated by social recognition. a** Matrices of interregional correlations derived from c-Fos-positive cells. Colors indicate the scale of Pearson coefficients (*r*) from 1 (red) to −1 (indigo). **b** Network graphs of significantly positive (*r* > 0.6, red lines) or negative (*r* < −0.6, indigo lines) correlations. **c** Pie charts of relative proportions of positive (red) and negative (indigo) correlations. **d** A box-and-whisker plot of average *r* in calculation of all interregional correlation coefficients. Matrices of interregional correlations for subregions in the medial prefrontal cortex (**e**), anterior cingulate cortex (**f**), hippocampus (**g**), and amygdala (**h**), together with box-and-whisker plots of average *r* for each region. The horizontal line inside each box indicated the median, the bounds of the box indicated the 25th and 75th percentiles, and the lower and upper whiskers indicated the 10th and 90th percentiles, respectively. *$p < 0.05$, two-tailed Fisher's LSD test. *ns*, not significant. *n* = 6 mice/group. Source data are provided as a Source Data file. ACC anterior cingulate cortex, Acg anterior cingulate cortex (rostral), BLA basolateral amygdala, BMP basomedial amygdala, CA1 hippocampus CA1, CA2/3 hippocampus CA2/3, CeA central nucleus of amygdala, DG dentate gyrus, dSTR dorsal striatum, Ect ectorhinal cortex, IL infralimbic cortex, lEnt lateral entorhinal cortex, M1 primary motor cortex, mHA medial hypothalamus, NA nucleus accumbens, OFC orbitofrontal cortex, PL prelimbic cortex, PRh perirhinal cortex, PtA parietal association cortex, RSD retrosplenial dysgranular cortex, RSG retrosplenial granular cortex, TeA temporal association cortex, VTA ventral tegmental area, vTH ventral thalamus.

the amygdala (BLA and BMP), PtA, lEnt, Acg, and VTA had high ranks for *Z*-scores, but their participation coefficients were zero (Fig. 5a–c, right). Lastly, we simulated the spatial distribution of the degree and betweenness centralities in a force atlas format, where distinct modules were color-coded. The control group contained two interactive modules (Fig. 5d). One module (pink) was centered on the hippocampus and mPFC, which critically regulate mnemonic processes[48]. The other module (green) included the parahippocampal cortices and CA2/3, which integrate sensory information for memory establishment[26,54]. Perturbation of the vermis destroyed these modules and produced solitary, amygdala-centered networks (Fig. 5e, f). Given that dynamic interactions among the amygdala, hippocampus, parahippocampus and mPFC are crucial for emotion-based memory[57,58], we propose that the cerebellum supports SRM by promoting and coordinating functional connectivity of the corticolimbic system.

**Anatomical tracing of the fastigial nucleo-cortical connections**

The vermis influences the corticolimbic system through projections to the FN[17]. To track the FN output connections, we injected AAV1-hSyn-Cre unilaterally into the FN of floxed tdTomato (Ai9) mice to label FN neurons via Cre-LoxP recombination[59] (Fig. 6a). Since AAV1-hSyn-Cre has an anterograde transsynaptic property[60], this experiment allowed us to trace FN axons to their first- and second-order targets in the forebrain. We transduced neurons mainly in the FN while avoiding the interposed and dentate nuclei (Fig. 6b). As most fastigial axons decussate to innervate downstream nuclei[12,17], we followed their projections contralateral to the injection site in the 24 brain areas that were examined for c-Fos expression (Fig. 4).

In line with recent anatomical studies in mice[21,22], we found prominent labeling of somas and processes of neurons in the vTH (ventrolateral and ventromedial thalamus) across individual mice (Fig. 6c, Supplementary Fig. 5). Labeled neurons were also found in the anteromedial (AM), centrolateral (CL), centromedian (CM), mediodorsal (MD), and paraventricular (PV) thalamus (Fig. 6c, top and middle). This reinforces that the thalamus is a major efferent pathway for the FN to the neocortex[21,22]. Moreover, we observed neurons expressing tdTomato sparsely distributed in the dorsomedial and lateral hypothalamus (Fig. 6c, bottom). The cerebellar-hypothalamic contacts may be important for somatic and visceral integration[19,20]. In the midbrain, labeled axons (no somas) were present in the VTA, consistent with the previous report on a lack of monosynaptic connection between the FN and VTA[21]. Labeled neurons and their processes were found in the substantia nigra pars compacta (SNc) and red nucleus (RN), indicating direct FN projections to the basal ganglia and rubrospinal tract involved in motor and cognitive functions[7–9].

Among the second-order targets, axonal (not somatic) labeling was evident in subcortical regions, e.g., the dorsal striatum (dSTR) and nucleus accumbens (NAc) (Fig. 7b, Supplementary Fig. 5). But the labeling was scarce in the hippocampus and amygdala (Fig. 7d), confirming no mono- or disynaptic innervation from the FN to these areas[21]. The engagement of hippocampus and amygdala in cerebellum mediated SRM (Fig. 5) is likely through their intense crosstalk with other brain regions, such as the cerebral cortex[61,62]. In fact, we observed a high density of labeled axons in somatosensory (S1) and motor (M1) cortices, as well as in areas involved in decision-making, emotion and attention, e.g., orbitofrontal cortex (OFC), ACC, TeA, and PtA (Fig. 7a–c). There was moderate labeling in the mPFC (Acg and PL), retrosplenial regions (RSD and RSG), ectorhinal (Ect), perirhinal (PRh), and entorhinal (lEnt) cortices (Fig. 7a, c, e), which are crucial for episodic memory[54]. The IL was sparsely labeled here (Fig. 7a) but not in another study[21], possibly due to differences in injection sites or floxed mice (Ai9 vs Ai14). The abundance of labeled neurons and their processes was summarized for the 24 regions (Supplementary Fig. 5). Collectively, these data provide an anatomical basis for cerebellar regulation of SRM. However, the functional connectivity for each region derived from the c-Fos mapping (Figs. 3–5) does not simply correlate to its structural connection with the medial cerebellum (Figs. 6, 7), suggesting the complexity of the inter-nucleus circuits subserving SRM.

**Amygdalar integration in the fastigial nucleo-cortical circuits**

To understand how the amygdala, specifically BLA, becomes a central hub in the SRM network after vermal manipulation (Fig. 5), we combined anterograde tracing of the FN output with retrograde tracing of the BLA input by infusing AAV1-hSyn-Cre in the FN and AAVrg-hSyn-EGFP in the contralateral BLA of Ai9 mice (Fig. 8a). As shown in Figs. 6 and 7, AAV1-hSyn-Cre[59,60] permitted transduction of tdTomato in the FN (Fig. 8b) and its downstream neurons. Being a retrograde virus, AAVrg-hSyn-EGFP[63] labeled the BLA (Fig. 8c) and its upstream neurons with EGFP. Using the two fluorescent signals, we probed the anatomical relationship between the FN and BLA. In line with the literature[64], the BLA was connected with the cortex including auditory, visual, somatosensory, motor (M1), prefrontal (PL and IL), temporal (TeA), ectorhinal (Ect), perirhinal (PRh), and entorhinal (lEnt) cortices, as well as the subcortex including basal ganglia (dSTR) and thalamus (Supplementary Fig. 6). In the thalamus, we found a great number of EGFP-expressing neurons in the PV, parafascicular (PF), MD, and medial geniculate (MG) nuclei but rarely in the vTH (Fig. 8d, e), contrasting to the preferential target of the FN in the ventral part (Fig. 6c). The thalamic input to the BLA is known for conveying sensory information for fear learning and memory[64]. In the PV nucleus where both tdTomato- and EGFP-expressing somas were present, they did not overlap, implying that the FN projects to different neurons from those sending input to the BLA. In the midbrain, EGFP-labeled somas were found in the VTA while EGFP-labeled axons (not somas) were in the NAc (Fig. 8f, g). This represents a well-known VTA-to-BLA-to-NAc circuit involved in reward seeking behavior[65]. In the VTA and NAc, very few EGFP-labeled somas or axons contained tdTomato, the indicator for the FN projections. In the cortices that were identified as the second-order FN targets (Fig. 7), e.g., M1, OFC, PtA, Ect and PRh, we noticed EGFP-expressing somas, but they rarely made close contact with axons expressing tdTomato (Fig. 8h, i). The cerebral cortex-BLA interactions are critical for coordinating emotional and cognitive responses[57,58,64]. The separate signals in all brain regions suggest that the FN does not connect with the BLA through hierarchical neurotransmission. However,

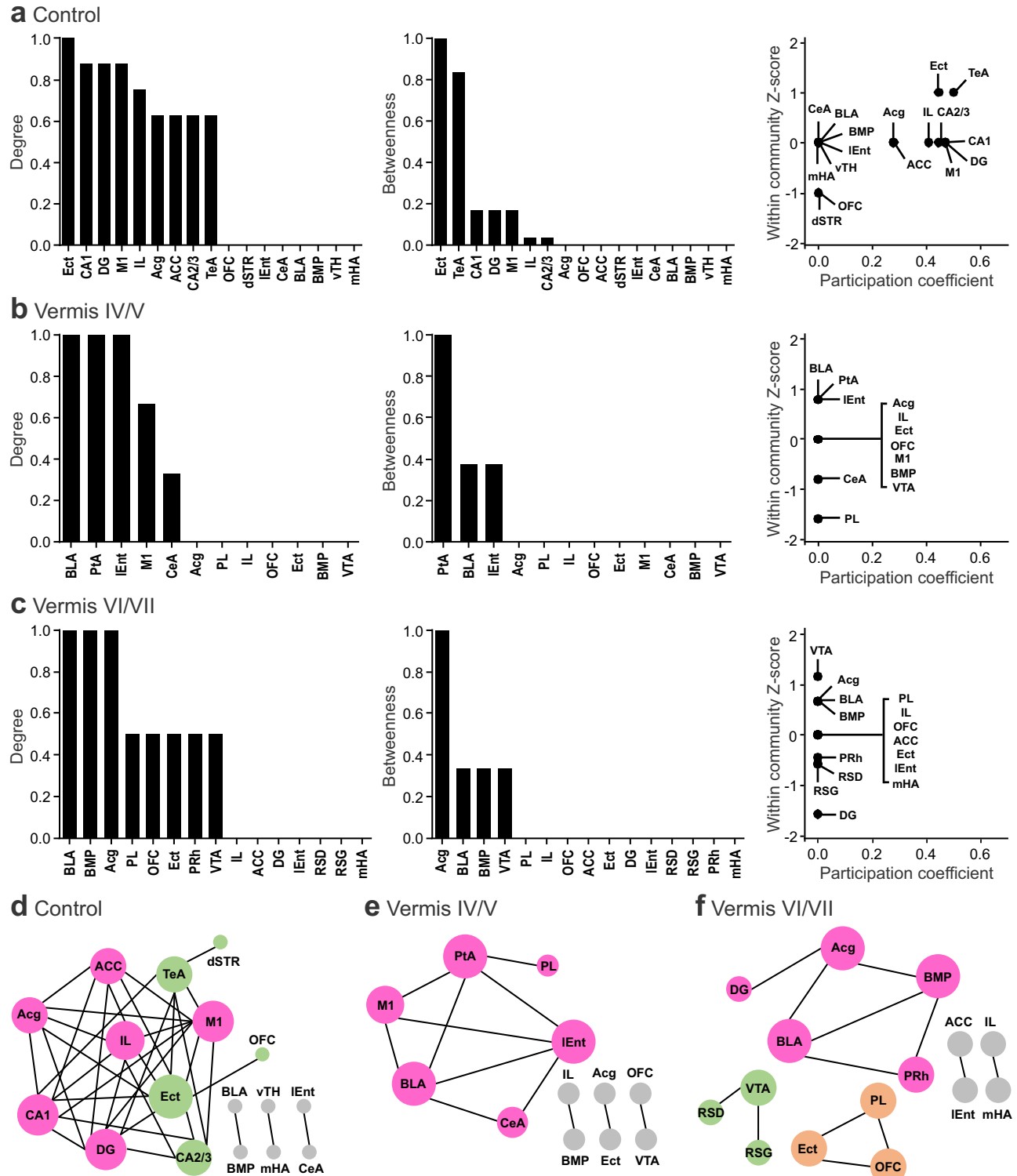

**Fig. 5 | Chemogenetic excitation of MLIs in cerebellar lobules IV-VII disrupted modular structure of social recognition network.** Rankings of normalized degree (left), betweenness (middle), within-community Z-scores and participation coefficients (right) of brain regions for control (**a**), vermis IV/V (**b**), and vermis VI/VII (**c**) groups. **d**–**f** Hubs revealed by graph theoretical analysis of neural networks activated during social recognition in all groups. Hubs were defined with modularity maximization, based on within-community Z-score and participation coefficient of each region. Distinct communities were color coded. Size of nodes (brain regions) was proportional to their degree. Note that participation coefficients for vermis IV/V and VI/VII groups were zero and thus they did not form >1 module.

Source data are provided as a Source Data file. ACC anterior cingulate cortex, Acg anterior cingulate cortex (rostral), BLA basolateral amygdala, BMP basomedial amygdala, CA1 hippocampus CA1 CA2/3, hippocampus CA2/3, CeA central nucleus of amygdala, DG dentate gyrus, dSTR dorsal striatum, Ect ectorhinal cortex, IL infralimbic cortex, lEnt lateral entorhinal cortex, M1 primary motor cortex, mHA medial hypothalamus, NAc nucleus accumbens, OFC orbitofrontal cortex, PL prelimbic cortex, PRh perirhinal cortex, PtA parietal association cortex, RSD retrosplenial dysgranular cortex, RSG retrosplenial granular cortex, TeA temporal association cortex, VTA ventral tegmental area, vTH ventral thalamus.

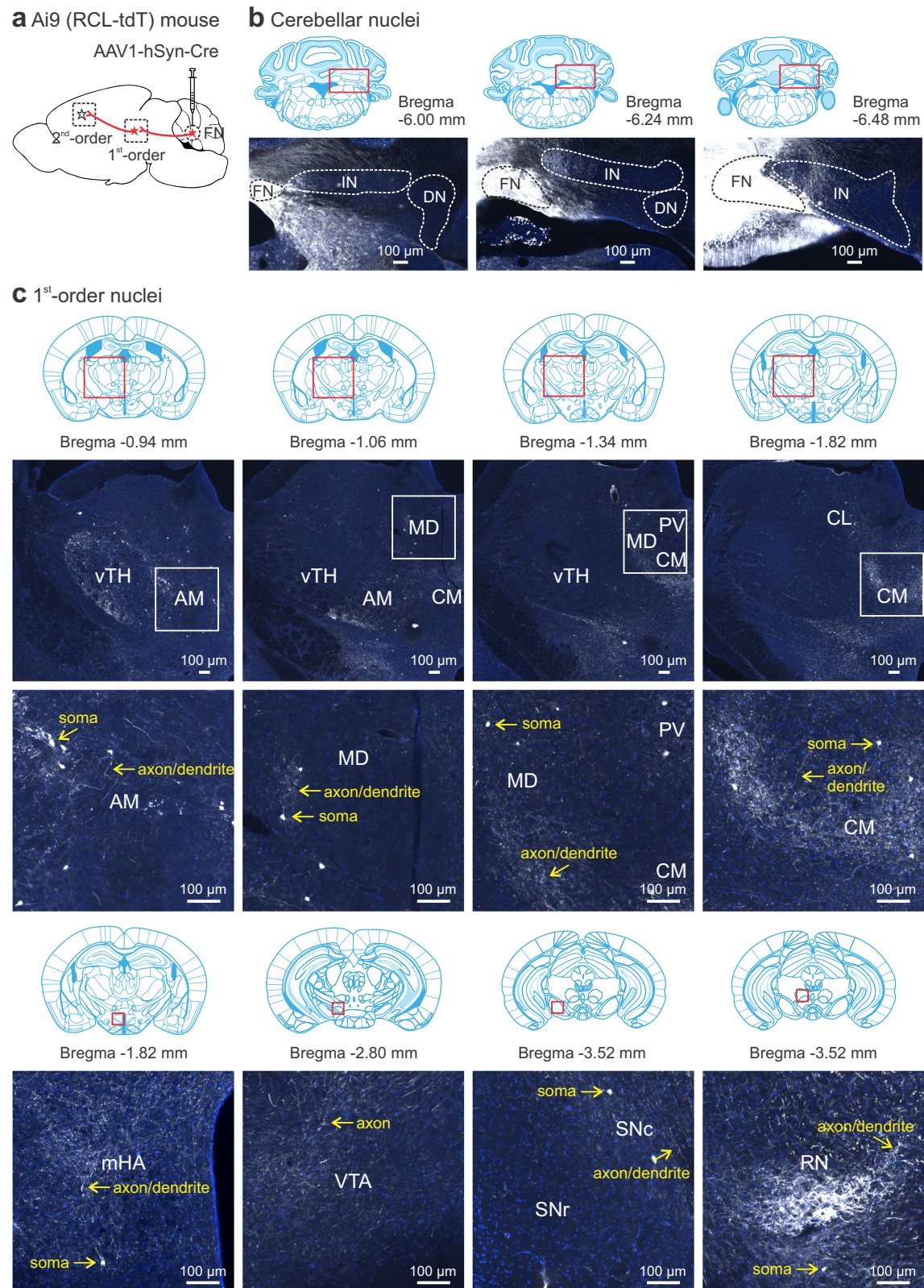

the coexistence of the FN output and BLA input within the same regions signifies a role of the BLA in integrating sensory and emotional information in the FN-neocortex circuits (Supplementary Fig. 7).

## Discussion

By manipulating cerebellar activity with spatiotemporal control, we have uncovered a new role for the cerebellum in incorporation of working memory into social behavior. Specifically, we found that activating MLIs in the vermis during memory recall impaired social, but not non-social, recognition memory. Neuroanatomical tracing and c-Fos functional mapping revealed the circuit basis for the cerebellum to co-activate and coordinate the limbic structures essential for emotional responses and cognitive functions in support of SRM. This work provides evidence for the internal forward model, in which the

**Fig. 6 | Transsynaptic tracing of cerebello-cortical circuits. a** Schematic of anterograde tracing by injecting AAV1-hSyn-Cre into the FN of an Ai9 mouse that has a LoxP site. Via Cre-LoxP recombination, transduced FN neurons express tdTomato and project to a first-order nucleus. As the virus has a transsynaptic property, it further transduces neurons in the first-order station, which send axons to a second-order nucleus. **b** Among cerebellar nuclei, the FN was primarily targeted as shown in three consecutive sections. Occasionally, the virus retrogradely transfected PCs in the cerebellar cortex. FN fastigial nucleus, IN interposed nucleus, DN dentate nucleus. **c** Examples of tdTomato labeling (in white) of somas, dendrites, and axons in the first-order nuclei (rostral to caudal): ventral thalamus (vTH),

medial hypothalamus (mHA), ventral tegmental area (VTA; few somas), substantia nigra pars compacta (SNc), substantia nigra pars reticulata (SNr), and red nucleus (RN). Thalamic subregions were highlighted in the middle panels, including anteromedial thalamus (AM), centrolateral thalamus (CL), centromedian thalamus (CM), mediodorsal thalamus (MD), and paraventricular thalamus (PV). The images represented average fluorescent intensity of 2-4 serial sections for each region in 3 mice. Cell nuclei were stained with DAPI (blue). All regions were outlined with the mouse brain atlas. The atlas was adapted from *Franklin, K. B. J. & Paxinos, G. The mouse brain in stereotaxic coordinates, compact third edition (Academic Press; 3rd edition, 2008).*

cerebellum conducts mental activity by orchestrating the cerebello-cortical network[13].

Although the cerebellum is known for motor coordination, our manipulations did not affect animals' motor performance on the accelerating rotarod or exploratory activity in less challenging conditions, e.g., in the open field (Supplementary Fig. 2). This can be for several reasons. First, our interference was restricted to a single lobule, as indicated by the localized hM3Dq expression (Fig. 1). Activation of MLIs in a lobule might not be sufficient to disturb general movement, even though it was sufficient to produce a SRM deficit. The sensitivity of the non-motor function to the cerebellar interference is congruent with other studies using similar approaches[23,24,37]. Second, we manipulated MLIs instead of PCs to attenuate, rather than silence, the output of the cerebellar cortex (Figs. 1, 2). This tactic limited potential alterations in motor skills, anxiety levels, social preference, and territorial aggression that might confound the assessment of SRM[45]. Lastly, the subregions that we perturbed are less involved in online motor control. While the anterior vermis may be part of the sensorimotor representation[14,15] and targeting lobule IV/V with photostimulation occasionally impeded exploratory behavior (Fig. 2g), the posterior vermis typically represents affective and cognitive functions[14,15] and interfering with lobule VI/VII had no effect on total exploration in the object and social recognition tests (Figs. 1, 2). Though these two tests had the same structure, vermis-perturbed animals were able to discriminate between objects by spending more time on the new object than the old one yet failed to show such a preference in identifying conspecifics. This demonstrates an explicit contribution of the cerebellum to SRM.

PCs fire APs spontaneously to encode output information of the cerebellar cortex to the neocortex. Reduced PC activity leads to motor and non-motor dysfunctions in neurodegenerative (e.g., ataxia) and neurodevelopmental (e.g., ASD) disorders[66]. It is difficult to pinpoint whether the extent of PC firing deficits, the lobule-specific targeting of genetic mutations, or the conditions of other brain regions result in certain behavior phenotypes. In the case of ASD, cerebellar abnormalities including PC loss and impaired PC activity and plasticity underlie a spectrum of behavioral deviations in humans and animal models[67,68]. However, it is noteworthy that restoring PC activity is to date the most effective avenue to rescue social deficits in ASD mouse lines[41–44]. APs elicited from PCs are driven by PC intrinsic excitability and regulated by MLI inhibition, both of which can alter PC firing rate and regularity[10]. Further analysis is needed to conclude whether global changes in speed or specific patterns of PC activity are crucial for processing social information.

By spatially and temporally tuning PCs, MLIs actively participate in learning and memory to define motor and non-motor functions[34–39]. For example, chemogenetic inhibition of MLIs in paravermal lobule VI or crus 1 causes deficient eyeblink conditioning responses[37]. The same manipulation in lobule VI affects reverse learning and in lobule VII affects novelty-seeking behavior. However, most of the effects are present in juvenile mice and diminish when animals reach maturity[37]. Our manipulation of MLIs in the adult mice also had little effect on learning, emotion, or sociability (Supplementary Fig. 2) except for SRM (Figs. 1, 2). This unexpected finding may be due to our experimental

design. Opposite to suppressing MLIs[37], we increased their activity based on a common phenotype among ASD mouse models[42–44]. Activation of the hM3Dq in the c-kit^IRES-Cre mice might have involved a small number of Golgi and glial cells in addition to MLIs[47] (Fig. 1). Yet, photostimulation of the nNOS-ChR2 mice solely excited MLIs[50] and produced the same SRM deficit as the chemogenetic manipulation (Fig. 2), arguing for the significant contribution of MLIs to SRM. Future investigation is required to determine how MLIs encode SRM with in vivo recordings of APs or Ca$^{2+}$ signals from MLIs during social recognition tasks.

SRM exploits working memory while judging a recently encountered conspecific[26–30]. The cerebellar involvement in working memory is well documented[12]. Schmahmann and Sherman first reported patients with damage to the posterior vermis displayed working memory loss along with other cognitive and affective symptoms[69]. Reduced volume or activity of the vermis is often present in neuropsychiatric disorders[17]. Animal work shows that interfering with PC activity in the vermis does not affect object recognition memory per se, like our data in Fig. 1 & 2, but compromises location- or context-based memory[24,25]. Also, the cerebellum participates in decision-making and consolidation of reward-related go/no-go memory tasks[70–72]. As social interactions are rewarding, the cerebellum may promote SRM via its connections to the reward pathway[73], which significantly overlaps with the social brain network[1–3].

We constructed the c-Fos-derived neural network underlying SRM and identified two main modules centering on the mPFC-hippocampus and Ect-TeA (Figs. 3–5). The mPFC-hippocampus module may be essential for processing episodic information of "what", "where" and "when"[48]; and the Ect-TeA module may be critical for integrating sensory information into the mPFC-hippocampus memory system[54]. Perturbing the cerebellum broke down the highly interactive clusters and shifted the brain network to an amygdala-centered module (Fig. 5). The emergence of amygdala as a hub reinforces our results that the vermis is of particular importance for emotion-related memory (Figs. 1, 2). As described before[17,19–22], the vermis and its projections to the FN sent output to diverse brain areas (Figs. 6, 7). The monosynaptic links to the limbic system (thalamus and hypothalamus) and brainstem, as well as disynaptic links with the prefrontal cortex, temporal cortex, parietal cortex, and striatum may specifically mediate the cognitive and affective processes in SRM[7–9]. However, we did not find mono- or disynaptic connections from the FN to the hippocampus and amygdala (Figs. 6, 7). Their functional connectivity with the vermis may be through the cerebral cortex[61,62]. A study proposes that the cerebellum regulates coherence of gamma oscillations between mPFC and hippocampus for spatial memory[74]. It is tempting to speculate that the cerebellum may impact hippocampal activity in a similar way to enable SRM. Our analysis of the amygdalar integration revealed coexistence yet nonoverlap of FN outputs and BLA inputs in the SRM network (Fig. 8, Supplementary Fig. 7). Likely via neuronal interactions within these regions, the vermis has influenced amygdalar activity and connectivity (Figs. 3–5). In general, direct synaptic contacts convey more information than indirect ones. However, the indirect FN-BLA connections may be crucial for processing complex information to modulate the effects of emotion on memory[57,58]. The fact that cerebellar

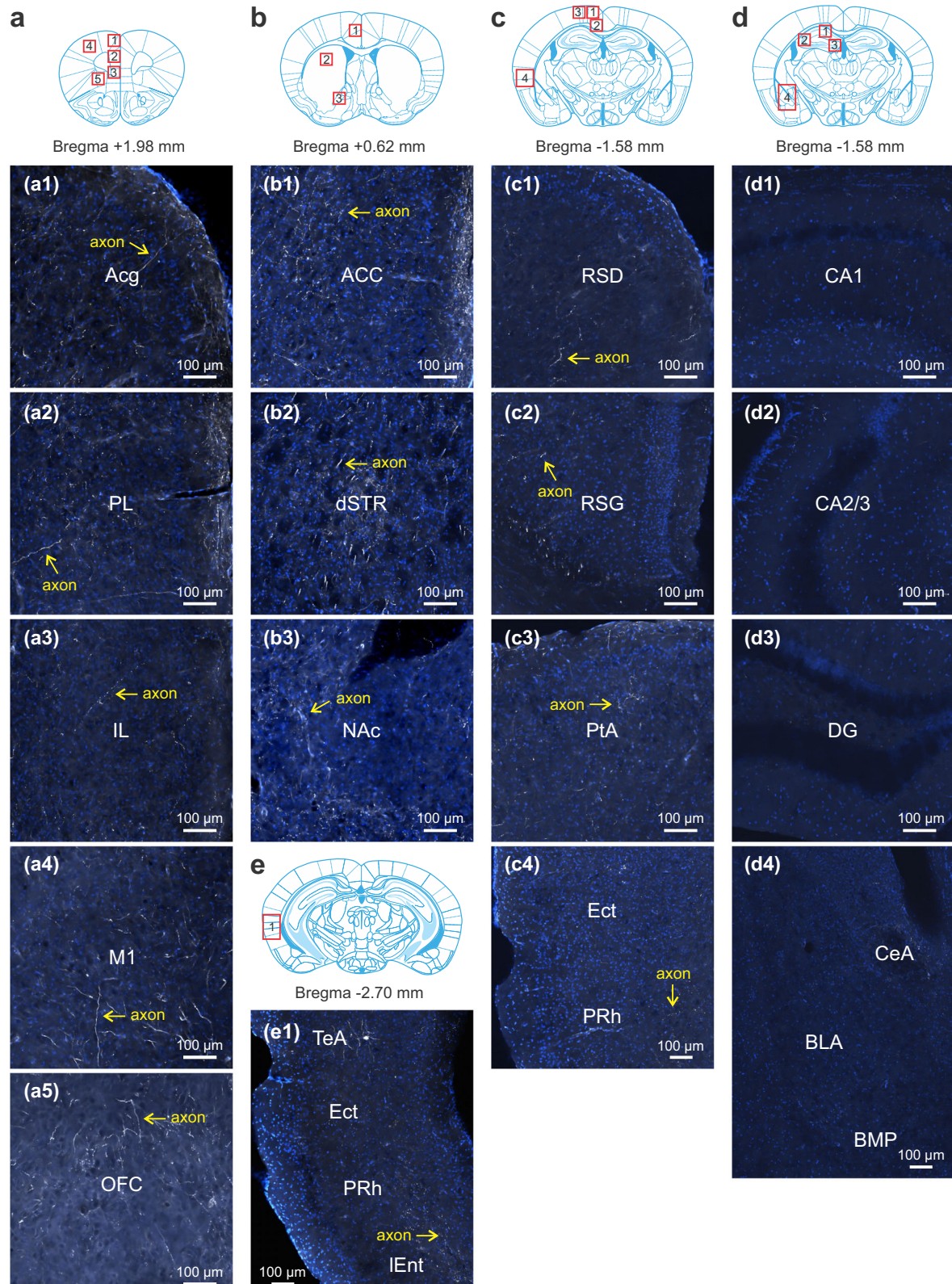

perturbation can alter these distal regions supports our view that the cerebellum is an integral part of the cerebello-cortical network for SRM.

Our data suggest that the cerebellum is necessary for retrieval, but not encoding, of SRM (Fig. 2). Memory encoding is the act of storing external features and internal states in a subset of neurons (engram) across various brain regions. Memory retrieval requires concurrent reactivation of the engram cells[49]. Here, we showed the SRM task primarily activated excitatory neurons in the local circuits that were responsible for sending information to other brain regions (Supplementary Fig. 4), providing a mechanism for the interregional coactivation of engram cells. The neuronal activity in the encoding and retrieval processes may differ. For instance, the hippocampal CA2/3 and DG are only active during memory formation while the subiculum

**Fig. 7 | Transsynaptic tracing of cerebello-cortical circuits (continued).**
**a**–**e** Examples of labeled axons and axon terminals (white) in the second-order nuclei that were connected to the fastigial nucleus via neurons in the first-order stations. Images were taken from cortical and subcortical areas in the same Ai9 mice (*n* = 3) infused with AAV1-hSyn-Cre into the fastigial nucleus as in Fig. 6. The cortical areas included Acg (a1), PL (a2), IL (a3), M1 (a4), OFC (a5), ACC (b1), RSD (c1), RSG (c2), PtA (c3), Ect (c4, e1), PRh (c4, e1), TeA (e1), and lEnt (e1). The sub-cortical areas included dSTR (b2), NAc (b3), CA1 (d1), CA2/3 (d2), DG (d3), CeA (d4), BLA (d4), and BMP (d4). The images represented average fluorescent intensity of 2–4 serial sections for each region in 3 mice. Cell nuclei were stained with DAPI

(blue). All regions were outlined with the mouse brain atlas. The atlas was adapted from *Franklin, K. B. J. & Paxinos, G. The mouse brain in stereotaxic coordinates, compact third edition (Academic Press; 3rd edition, 2008)*. ACC anterior cingulate cortex, Acg anterior cingulate cortex (rostral), BLA basolateral amygdala, BMP basomedial amygdala, CA1 hippocampus CA1, CA2/3 hippocampus CA2/3, CeA central nucleus of amygdala, DG dentate gyrus, dSTR dorsal striatum, Ect ectorhinal cortex, IL infralimbic cortex, lEnt lateral entorhinal cortex, M1 primary motor cortex, NAc nucleus accumbens, OFC orbitofrontal cortex, PL prelimbic cortex, PRh perirhinal cortex, PtA parietal association cortex, RSD retrosplenial dysgranular cortex, RSG retrosplenial granular cortex, TeA temporal association cortex.

(and CA1 to a less degree) is active during memory recall[75,76]. This selective activation in the initial learning may explain why c-Fos expression in the hippocampus was insensitive to the changes in later performance of memory retrieval (Supplementary Fig. 3). In addition to recollection of contextual cues, recognition memory entails familiarity for acquaintance-based discrimination[77]. While recollection involves the medial temporal lobe including parahippocampus and amygdala, familiarity involves the PRh. Remarkably, c-Fos levels in all the regions were elevated by the vermal perturbation that led to the SRM deficit (Fig. 3, Supplementary Fig. 3). This strengthens the specificity of the identified neural network for recognition memory. Our finding that interfering with the vermis decreased brain-wide functional connectivity (Fig. 4) indicates a unique position of the cerebellum in reinstatement of memory traces by coactivating the engrams.

The results of this study must be interpreted with caution due to several technical issues. First, the readouts of c-Fos expression have a modest temporal resolution. Despite being a validated tool[46], c-Fos expression is a slow transcriptional process and does not allow real-time monitoring of neural activity or resolve the differences between events occurring close in time. Thus, the c-Fos-based neural network may reflect interwound responses to encoding and recalling of SRM. Further inquiry using in vivo electrophysiology and Ca²⁺ imaging is needed to address how the engram cells are recruited and reactivated in a task-related manner with temporal accuracy. Second, the conceptualized network model of SRM inevitably embeds other information, such as the sensory and motor processing during the social recognition task as well as environmental factors, e.g., fear or stress from experimental handlings. While the cerebellum was taken for examining the distribution of fluorescent proteins and not used for c-Fos staining, c-Fos in the vermis can be reactive to external stimuli including odorants[78]. To minimize the variables, all groups were tested under the same conditions and c-Fos counts were normalized to the average value of the control group for each brain region (Figs. 3–5). Thereby, the network disruption may largely result from the cerebellar interference. Lastly, it is worth noting that the chemo- and optogenetic manipulations are artificial, from which we can infer what the cerebellum can do but not exactly what it does physiologically.

Notwithstanding these considerations, our study presents the first evidence for a causal link of the cerebellum to social memory, which advances our understanding of the neural substrates for social behavior. As social impairments are commonly associated with neuropsychiatric disorders[4–6], our results will help develop diagnostic and therapeutic strategies by targeting cerebellar modulation of the higher-order functions with cell-type- and region-specific precision.

## Methods

### Subjects

C-kit^IRES-Cre mice and nNOS-ChR2 BAC mice were provided by Hiroki Taniguchi (Ohio State University, USA)[47] and George J Augustine (Nanyang Technological University, Singapore)[50], respectively. Ai9 (RCL-tdT) mice (strain # 007909) were purchased from Jackson Laboratory (Bar Harbor, ME)[59]. Both sexes (8-12 weeks old) were included, and no sex differences were observed. Mice were kept under

a 12-hour light-dark cycle (light on from 07:00 to 19:00) and reared 3-5 per cage with food and water *ad libitum*. All procedures were approved by the Institutional Animal Care and Use Committee (IACUC) and the Institutional Biosafety Committee (IBC) of University of Minnesota, in accordance with the National Institutes of Health guidelines. Heterozygous c-kit^IRES-Cre were bred to obtain mice with or without (control) Cre recombinase, genotyped with 2 primers: 5′-CGGTCGATGCAACGA GTGATG-3′ and 5′-AGCCTGTTTTGCACGTTCACC-3′. Male nNOS-ChR2 were crossed with female C57BL/6 J to obtain wild-type (control) or nNOS-ChR2 mice, genotyped with 2 primers: 5′-AGTAGCTCAGGT TCCTGTGGG-3′ and 5′-GCAAGGTAGAGCATAGAGGG-3′.

### Stereotactic surgery

c-kit^IRES-Cre, nNOS-ChR2 and their littermate controls were randomly assigned to vermis IV/V and VI/VII surgical groups. Animals were anesthetized with a cocktail of ketamine and xylazine (100 and 10 mg/kg, respectively, i.p.). Carprofen (5 mg/kg, s.c.) was administered as an analgesic. Mice were fixed on a stereotactic frame (Stoelting, Wood Dale, IL) and an incision was made to expose the skull. For viral infusion, a hole was drilled above the cerebellum of c-kit^IRES-Cre mice and their wild-type littermates according to the coordinates: AP -6.0 or -7.0 mm (for vermis IV/V or VI/VII, respectively), ML ± 0.0 mm, relative to bregma[79]. AAV8-hSyn-DIO-hM3Dq-mCherry (titer>3.0 × 10¹³ vg/ml, 0.1–0.15 μl, University of Minnesota Viral Vector and Cloning Core) was infused into the cerebellum (DV −1.0 mm) with a 10 μl Hamilton syringe controlled by a microinjection pump (QSI 53311, Stoelting) at a flow rate of 0.09 μl/min. After infusion, the needle was left in place for >7 min and then slowly retracted. The wound was sutured and disinfected with 70% ethanol and iodine. For optic-fiber implantation, a hole was drilled above the cerebellum of nNOS-ChR2 mice and their wild-type littermates according to the coordinates: AP −6.4 or −7.0 mm (for vermis IV/V or VI/VII, respectively), ML ± 0.0 mm, relative to bregma[79]. An optic fiber (∅200 μm) attached with a ceramic ferrule (FT200EMT and CFLC230-10, Thorlabs, Newton, NJ) was implanted in the hole 0.0–0.2 mm below the cerebellum surface. Two screws and dental cement were applied to secure the optic fiber on the skull. For neuronal tracing, AAV1-hSyn-Cre (plasmid #105553, titer >1 × 10¹³ vg/ml, Addgene, Watertown, MA) was unilaterally infused into the FN of Ai9 (RCL-tdT) mice according to the coordinates: AP −6.24 mm, ML 0.7 mm, DV −3.3 mm, relative to bregma[79]. AAVrg-hSyn-EGFP (plasmid #50465, titer ≥7 × 10¹² vg/ml, Addgene) was infused into the BLA according to the coordinates: AP −1.34 mm, ML 3.0 mm, DV −4.6 mm, relative to bregma[79]. After operations, mice were placed on a heating pad (37 °C) to maintain body temperature until awake. Electrophysiology and behavioral tests were conducted 3 weeks after surgeries. Imaging of labeled neurons was done 4–5 weeks after surgeries. Mice (*n* = 1–2/group) were excluded when injections were outside the intended areas.

### Electrophysiology

c-kit^IRES-Cre mice (*n* = 6) and their wild-type littermates (*n* = 7) injected with AAV8-hSyn-DIO-hM3Dq-mCherry and nNOS-ChR2 mice (*n* = 3) were used for recordings. Following decapitation, the brain was immediately dissected, and sagittal cerebellar slices were sectioned at

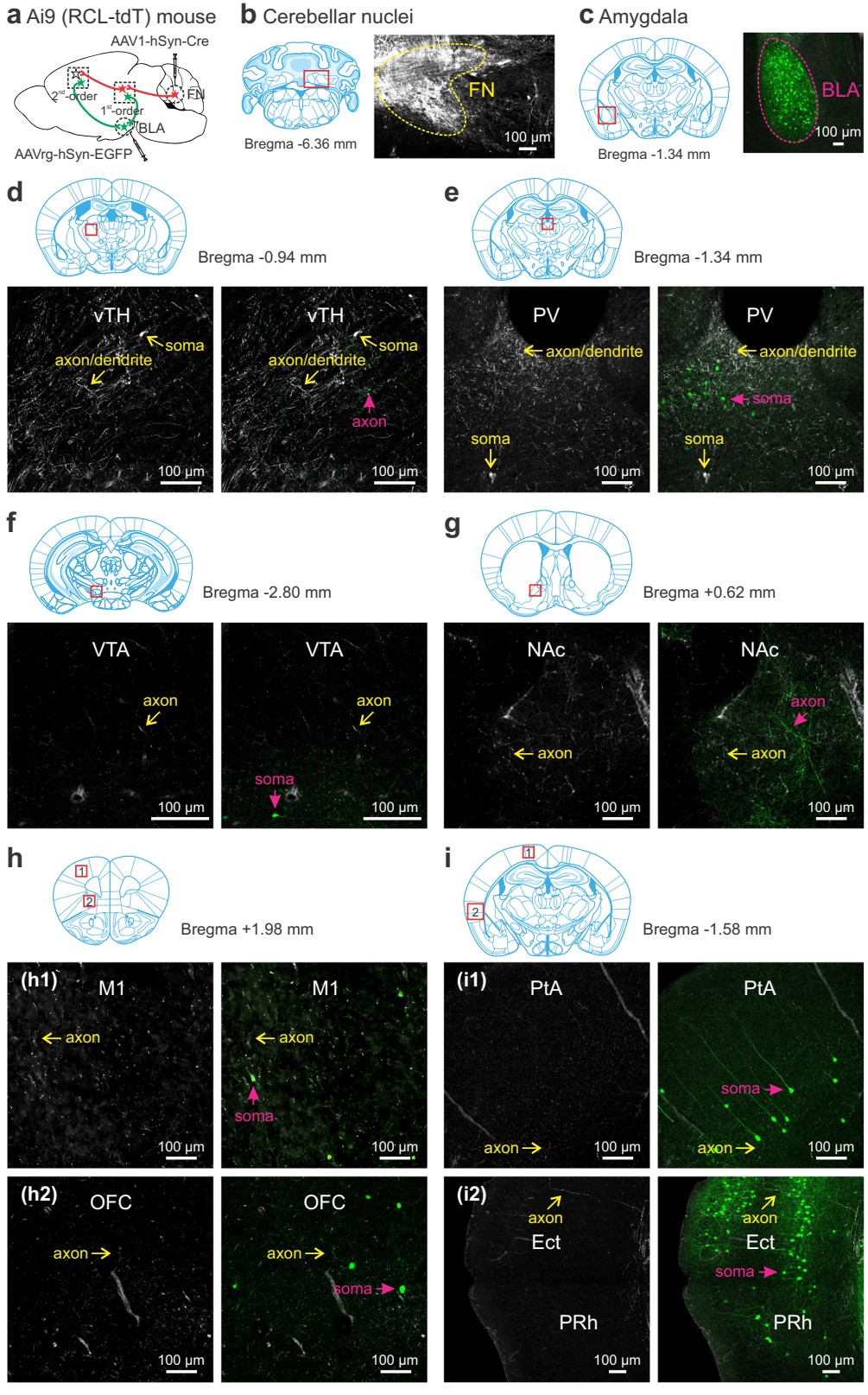

Nature Communications | (2023) 14:6007

a thickness of 300 μm using a vibratome (VT1200S, Leica Biosystems, Buffalo Grove, IL) in ice-cold modified artificial cerebrospinal fluid (ACSF). It contained (in mM): sucrose (217.6), KCl (3), glucose (10), $NaH_2PO_4$ (2.5), $NaHCO_3$ (26), $MgCl_2$ (2), and $CaCl_2$ (1), continuously bubbled in 95% $O_2$ and 5% $CO_2$ (pH 7.4). Slices were then incubated in oxygenated standard ACSF including (in mM): NaCl (125), KCl (2.5), glucose (10), $NaH_2PO_4$ (1.25), sodium pyruvate (2), myo-inositol (3),

ascorbic acid (0.5), $NaHCO_3$ (26), $MgCl_2$ (1), and $CaCl_2$ (2) (pH 7.4) at 37 °C for 30 min prior to experimentation. Slices were transferred to a recording chamber mounted on a BX51WIF Olympus microscope with a 60x water immersion objective. They were perfused with the standard ACSF at a rate of ~1 ml/min. PCs and MLIs were identified by their size and location. Patch electrodes had resistances of 2.5-3 and 4.5-6 MΩ for PCs and MLIs, respectively. APs at the soma were

**Fig. 8 | Simultaneous tracing of the FN outputs and BLA inputs. a** Strategy for anterograde tracing of the fastigial nucleus (FN) outputs with AAV1-hSyn-Cre and retrograde tracing of the basolateral amygdala (BLA) inputs with AAVrg-hSyn-EGFP in Ai9 mice (*n* = 3). FN and its downstream neurons were tagged with tdTomato as shown in Fig. 6 & 7. BLA and its upstream neurons were labeled with EGFP. Examples of viral expression in the FN (white; **b**) and BLA (green; **c**), contoured with dotted lines to show the transduced areas. **d**–**i** Images taken from subcortical (**d**–**g**) and cortical (**h**–**i**: h1, M1; h2, OFC; i1, PtA; i2, Ect and PRh) regions showing the FN downstream neurons and their processes (white; indicated with yellow arrows), and BLA upstream neurons (green; indicated with magenta arrows). Note that the white and green signals rarely overlapped. In the vTH (**d**) and NAc (**g**), green axons (not somas) were present, implying these areas are the BLA output targets rather than provide input to the BLA. The images represented average fluorescent intensity of 2–4 serial sections for each region in 3 mice. All regions were outlined with the mouse brain atlas. The atlas was adapted from *Franklin, K. B. J. & Paxinos, G. The mouse brain in stereotaxic coordinates, compact third edition (Academic Press; 3rd edition, 2008)*. Ect ectorhinal cortex, M1 primary motor cortex, NAc nucleus accumbens, OFC orbitofrontal cortex, PRh perirhinal cortex, PtA parietal association cortex, PV paraventricular thalamus, VTA ventral tegmental area, vTH ventral thalamus.

recorded in the cell-attached mode with GΩ seal at −60 mV for PCs and −70 mV for MLIs. Intracellular solution contained (in mM): K-gluconate (97.5), KCl (32.5), EGTA (0.1), HEPES (40), MgCl$_2$ (1), ATP (2), GTP (0.5) (pH 7.3). All recordings were acquired on-line at 35 °C, filtered at 4 kHz, digitized at 50 kHz with a dual-channel amplifier (MultiClamp 700B, Molecular Devices, San Jose, CA) and digitizer (Digidata 1550B, Molecular Devices). Data were analyzed off-line with MiniAnalysis 6.0.7 (Synaptosoft, Fort Lee, NJ), Clampfit 10 (Molecular Devices) and Excel 2016 (Microsoft, Redmond, WA). To activate hM3Dq receptor, CNO (BML-NS105, Enzo Life Sciences, Farmingdale, NY) was applied to slices[43]. To excite ChR2, 470 nm LED light (LEDD1B, Thorlabs) was delivered through the 60x objective coupled with a GFP filter cube (Olympus)[50]. Light pulses were controlled using pCLAMP 10 (Molecular Devices). Light intensity (maximum = 8.2 mW/mm$^2$ at the tissue) was measured with a PM100D power meter and a S121C photodiode sensor (Thorlabs).

**Behavioral testing**
Either in chemogenetic (using c-kit$^{IRES-Cre}$) or in optogenetic (using nNOS-ChR2) experiments, three groups (*n* = 6–10 mice with 3–5 females/group) were included: control, vermis IV/V, and vermis VI/VII. For chemogenetic experiments, CNO was administered in all groups (1 mg/kg, i.p.) 30–40 min prior to each test. CNO was stored in aliquots at −20 °C. It was diluted in saline, kept on ice, and protected from light while in use. For optogenetic experiments, only open field, object recognition and social recognition tests were conducted. Light pulses were generated with a Polygon DMD pattern illuminator (Mightex, Toronto, Canada) powered by 470 nm LED through a cable connected to the optic fiber implanted on the animal's head. Light intensity (maximum=5.9 mW/mm$^2$ at the tissue) was measured with a PM100D power meter and a S121C photodiode sensor (Thorlabs).

**Apparatus.** An elevated plus maze consisted of a central platform (5 × 5 cm), two walled arms (25 × 5 × 25 cm) and two arms without walls (25 × 5 cm) in the shape of a cross. It was elevated 30 cm high, and the two types of arms were situated opposite to each other. A digital rotarod (Rotamex, Columbus Instruments, Columbus, OH) was used to test motor coordination. A box composed of 3 chambers (20 × 40 x 30 cm each) with passages (5 × 5 cm) within the walls that divided the chambers was used for three-chamber social tests. A box (40 × 40 × 30 cm) made of polyvinyl chloride was used for open field, object recognition, social recognition, and reciprocal social interaction tests. Different sets of objects with variable textures (smooth or rough), sizes (diameter 7–9 cm, height 14–17 cm) and shapes (column, irregular) were used for object recognition. Objects were weighed enough to prevent being moved by animals. Assignment of objects was counterbalanced to minimize object and/or place preference. Age-, sex-, and strain-matched wild-type mice served as strangers for social tests. Metal grid cups (diameter 10 cm, height 12 cm) were used to constrain strangers. Placement of strangers was counterbalanced between subjects. All tests were done in a quiet room (< 40 dB) under dim light (~50 lx) during 10:00-16:00. Test designs were based on previous studies using similar experimental protocols[25,43,44]. Apparatus was cleaned with 70% ethanol between subjects. A camera was connected to a computer to record and analyze behavior via tracking software (ANY-maze, Stoelting). Object and social exploration were manually counted by experimenters who were blind to the design. Other measurements were automatically made with the ANY-maze software.

**Elevated plus maze.** This test was used to assess anxiety-like behavior. A subject was placed onto the central platform (facing an open arm) and allowed to freely explore for 5 min. Entries to the center, open and closed arms, time spent in these areas, and time spent on head-dip and body-extension were analyzed.

**Open field.** This test was used to assess locomotor and exploratory activity. A subject was put into the center of the open field and monitored for 15 min. Traveling distance and duration, rearing, grooming, thigmotaxis, and center stay (virtual central square 13.3 × 13.3 cm) were analyzed in 5 min time bins.

**Rotarod.** This test was used to assess motor coordination and motor learning. A subject was placed on a rotarod rotating at a speed of 4 rpm until it could stay on for >1 min. Then, the speed accelerated from 4 to 40 rpm in 5 min, which was taken as one trial. Three trials were conducted with an inter-trial interval of 15 min. Rotating speed and time staying on the rotarod before falling were recorded.

**Reciprocal social interaction.** This test was used to assess pro-social behavior in a native situation without physical barriers. A subject was placed into the open field along with a stranger for 10 min. An interaction was defined as the animal' head being in physical contact with any body part of the other's, except for the tail. Duration of physical contacts was recorded.

**Three-chamber social test.** This test was used to assess social and social novelty preferences. It consisted of three sessions: habituation, sociability, and social novelty (9 min each). In the habituation trial, a subject was placed into the middle chamber and allowed to freely explore the whole apparatus. In the sociability trial, a stranger that had never been contacted by the subject was put underneath the cup in one of the side chambers. An identical empty cup was placed on the other side. In the social novelty trial, another stranger was put underneath the previously empty cup. Physical contact around the cups by the subject's nose, head and forelimbs was defined as explorative behavior. Sociable index = [time for exploring the stranger mouse − time for exploring the empty cup] / total exploration time. Social novelty index = [time for exploring the novel stranger − time for exploring the old stranger] / total exploration time. Positive values indicate intact social and social novelty preferences. Animals that showed a bias for either chamber (stayed >7 min) in the habituation trial were excluded (*n* = 1–2/group).

**Social recognition.** This test was used to assess social recognition memory. It included a learning trial (7 min) and a testing trial (7 min) with an inter-trial interval of 45 min. In the learning trial, a stranger was put underneath a cup on one side of the open field and an identical

empty cup was placed on the other side. In the testing trial, another stranger was put underneath the previously empty cup. Physical contact around the cups by the subject's nose, head and forelimbs was defined as explorative behavior. Sociable index = [time for exploring the stranger mouse − time for exploring the empty cup] / total exploration time. Social novelty index = [time for exploring the novel stranger − time for exploring the old stranger] / total exploration time. Positive values indicate intact sociability and social memory[29]. Animals that did not explore all presented stimuli in either trial were excluded ($n = 1$–2/group).

**Object recognition.** This test was used to assess object-based recognition memory. It included a learning trial and a testing trial with an inter-trial interval of 45 min. In the learning trial, two distinct objects were placed in the open field and a subject freely explored them for 7 min. In the testing trial, one of the objects was replaced with a novel object and the subject explored them for 7 min. Object exploration was defined as physical contact with the objects by the head, nose, or forelimbs, but not other behaviors, such as climbing the objects or sitting next to them. To minimize individual differences, the index = [time for exploring the novel object − time for exploring the old object]/total exploration time was calculated. A positive value indicates intact object memory. Animals that did not explore all presented stimuli in either trial were excluded ($n = 1$-2/group).

### Histology, Immunohistochemistry, and Imaging
Mice were deeply anesthetized and perfused with ice-cold phosphate-buffered saline (PBS) followed by 4% paraformaldehyde. The brain was removed, immersed in 4% paraformaldehyde at 4 °C overnight, transferred into 30% sucrose, and stored at 4 °C until processed. The brain was sectioned into 40–50 μm-thick slices with a microtome (SM2010R, Leica Biosystems). The cerebellum was taken for examining the distribution of fluorescent proteins and not used for c-Fos immunostaining.

**c-Fos.** Same batch of c-kit[IRES-Cre] and their wild-type littermates that underwent behavioral testing were used for c-Fos staining ($n = 6$ mice/group). 90 min after the social recognition test, mice were sacrificed, and their brains were fixed. Brain sections were pretreated with 0.3% $H_2O_2$ for 10 min and incubated in a blocking solution (2% goat serum and 0.2% Triton X-100 in PBS) at 37 °C for 30 min. Sections were then incubated with a rabbit anti-c-Fos antibody (1:4000, ab190289, Abcam, Cambridge, MA) at 4 °C for 24–48 h. Sections were washed with 0.2% Triton X-100 and incubated with biotinylated anti-rabbit IgG (1:200, BA-1000, Vector Labs, Burlingame, CA) at room temperature for 2 h. After another rinse with 0.2% Triton X-100, sections were incubated in ABC Kit (Vector Labs) for 1 h. 3,3' diaminobenzidine chromogen was used for visualization. Sections were mounted on slides, cover-slipped with a mounting medium (H-5000, Vector Labs) and imaged with a Leica DMi8 microscope under a 10x lens. 24 brain regions were identified based on the mouse brain atlas[79]. The number of c-Fos-positive cells in each region was counted bilaterally across 2-4 serial sections with a 200 μm interval. Region of interest (ROI) was adjusted to the size of each brain region and the same ROI was applied to the same region between groups. The number of c-Fos-positive cells in each group was divided by the average value of the control group and presented as fold change. Experimenters who were blind to the design counted labeled cells with semi-automated analysis by ImageJ (NIH).

**Calbindin, CaMKII, and GAD67.** To mark PCs, brain sections were incubated with a rabbit anti-calbindin D28K antibody (1:1000, PA1-931, Invitrogen, Waltham, MA) at 4 °C for 24–48 h. After being washed in PBS, sections were incubated with a goat anti-rabbit Cy5 (1:500, A10523, Invitrogen) at room temperature for 3 h. After another rinse

with PBS, sections were mounted on slides and cover-slipped with a mounting medium (H-1500, Vector Labs). To identify subtypes of c-Fos positive neurons following the social recognition test, sections were incubated with primary antibodies of guinea pig anti-c-Fos (1:1000, 226308, Synaptic Systems, Gottingen, Germany), rabbit anti-CaMKII (1:200, PA5-99558, Invitrogen) and mouse anti-GAD67 (1:500, MAB5406, Millipore, Burlington, MA) at 4 °C for 24–48 h. After being washed in PBS, sections were incubated with secondary antibodies of goat anti-guinea pig Alexa 488 (1:1000, A11073, Invitrogen), goat anti-rabbit Alexa 555 (1:1000, A21428, Invitrogen) and goat anti-mouse Alexa 647 (1:1000, A21236, Invitrogen) at room temperature for 3 h. After another rinse with PBS, sections were mounted on slides and cover-slipped with a mounting medium (H-1500, Vector Labs). Images were taken with a Zeiss LSM 710 confocal microscope and Zen 2.0 software (Oberkochen, Germany).

**Neuronal tracing.** Ai9 (RCL-tdT) mice ($n = 3$) infused with AAV1-hSyn-Cre in the FN and AAVrg-hSyn-EGFP in the BLA were used for anatomical tracing. Via Cre-LoxP recombination, neurons in the FN were labeled with tdTomato[59]. As AAV1-hSyn-Cre has a transsynaptic property[60], it further transduced FN downstream neurons. Example images from such a downstream region (PL) was shown in Supplementary Fig. 8. AAVrg-hSyn-EGFP is a retrograde virus[63]. It targeted the BLA and its upstream neurons. 4-5 weeks after viral infusion, brain sections were collected in series with a 200 μm interval, mounted on slides and cover-slipped with a mounting medium (H-1500, Vector Labs). Images were taken with a Zeiss LSM 710 confocal microscope and Zen 2.0 software (Oberkochen, Germany). Location of somas, dendrites and axons of tdTomato- and EGFP-expressing neurons was identified based on the mouse brain atlas[79].

### Interregional c-Fos correlations and graph theoretical analysis
Matrices of pairwise correlations were created by calculating Pearson coefficients ($r$) from the numbers of c-Fos-positive cells interregionally within each group ($n = 6$ mice). $r$ was Fisher $Z$ transformed. Statistics was conducted to compare average $r$ between groups. Positive and negative $r$ were indicated in red and indigo respectively in a gradient spectrum. Each node represented a brain region, and each connecting line represented Pearson's $r$ between regions. Graph theoretical analysis was used to illustrate network properties for social memory. The network was constructed with unweighted adjacency matrices for each group based on significant (uncorrected $p < 0.05$) and positive ($r > 0.6$) interregional correlations of c-Fos counts. Clusters were identified through enumeration of all potential community structures and optimization of the modularity. Degree (the number of edges that a node has) and betweenness (the number of shortest paths that pass through a node) were calculated to reveal the centrality of a node in the network. Within-community connectivity ($Z$-score, the extent of a node connected in its own community) and between-community connectivity (participation coefficient, the extent of a node connected to other communities) were calculated to reveal the modularity of the network. Graph theoretical analysis was performed in MATLAB (R2015b) with Brain Connectivity Toolbox (https://sites.google.com/site/bctnet/). Calculation details were described previously[55,56].

### Statistics
One-way or mixed two-way ANOVA with a between factor ("group") and a within factor ("object", "interval" or "trial") was applied to analyze data. Paired or unpaired $t$-tests were used when main effects were found. One sample $t$-test was used to compare the indices to 0 (chance level). Fisher's LSD was applied to *post hoc* tests when appropriate. Sample sizes were based on previous studies using similar experimental protocols[25,43,44]. For behavioral tests, $n$ denoted the number of mice. For other analyses, $n$ denoted the number of cells or samples

from >3 mice/group. All measurements were taken from distinct samples. $n$ values were given in the Source Data file and figure legends. All statistics were two-tailed tests with significance set as $p < 0.05$ using IBM SPSS statistics 25 (IBM Corp, Armonk, NY) and GraphPad Prism 9.3.1 (GraphPad Software, San Diego, CA). Data were presented as the mean ± standard error of the mean (SEM). Statistical details including $p$ values and degrees of freedom were summarized in Supplementary Table 1.

### Reporting summary

Further information on research design is available in the Nature Portfolio Reporting Summary linked to this article.

## Data availability

All data are available in the main text and supplementary information. Source data are provided with this paper and deposited to the Figshare (https://doi.org/10.6084/m9.figshare.22647700). The mouse brain atlas used in this study is publicly available[79].

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

## Acknowledgements

This study was supported by the National Institutes of Health (NIH) grants R15 NS112964 and R01 MH129300 to Y.M.Y. and R01 NS112289 to J.M.C., the Singapore Ministry of Education grant MOE2017-T3-1-002 to G.J.A., the Brain & Behavior Research Foundation Young Investigator grant to O.Y.C., and the Winston and Maxine Wallin Neuroscience Discovery Fund to YMY. AAV8-hSyn-DIO-hM3Dq-mCherry was generated by Dr. Ezequiel Marron Fernandez de Velasco in the University of Minnesota Viral Vector and Cloning Core. AAV1-hSyn-Cre and AAVrg-hSyn-EGFP were purchased from Addgene (gifts from Drs. James M Wilson and Bryan Roth). We thank the Structural Circuits Core (University of Minnesota) for providing technical support and Dr. Pei Wern Chin (Nanyang Technological University) for critical input.

## Author contributions

O.Y.C. and Y.M.Y. designed the project, performed the experiments, and analyzed the data with assistance from S.S.P. and HZ. G.J.A., J.M.C., C.K. and H.T. provided the mouse lines and technical input. O.Y.C. and Y.M.Y. wrote an early version of the manuscript and all other authors contributed to the final edition.

## Competing interests

The authors declare no competing interests.
