## [Peer Review File · Nature Communications]

Social memory deficit caused by dysregulation of the cerebellar vermisREVIEWER COMMENTS

Reviewer #1 (Remarks to the Author):

The manuscript by Chao and co-workers describes disruption of social recognition memory (SRM) by chemogenetic or optogenetic excitation of cerebellar molecular layer interneurons (MLIs) in lobules VI and VII. In addition, the authors present a cFos measurement of activity in different brain regions involved in the output of the cerebellum and make the interesting finding that chemogenetic excitation of Vermis VI/VII decreases interactions with the corticolimbic system. These results are novel providing a circuit basis for social impairment in autism and other psychiatric disorders and contributes to emerging evidence indicating that the cerebellum is involved in non-motor brain processes. In my opinion this is a solid and interesting manuscript. However, I find problems with data presentation, and lack of a thorough analysis of first order and second order DCN projections in Figure 6.

Major comments:

1. The analysis of DCN cerebellar projections in Figure 6 is disappointing. Please provide a more thorough analysis and a summary diagram. How do these findings relate to published findings by other investigators?
2. The presentation of the data is uneven making it difficult for the reader to evaluate the conclusions. First, some of the figures show individual data points (e.g. Fig. 1e,f,g) while other figures do not (e.g. Fig. 3b,c). Please show individual points in all figures. Second, the bar graphs show the standard error of the mean that does not provide information on whether there are statistical differences. A better measure is to show the bootstrapped 95% confidence interval (Hasley et al 2015 <https://doi.org/10.1038/nmeth.3288>) or the standard deviation.
3. For Figure 4 please provide examples of the graphs showing the correlation of cFos expression between different brain areas that was used to calculate the Pearson correlation coefficients. Please state the number of mice per group. Is this analysis appropriately powered? Did the cFos measurements include cFos measurement in DCN and the cerebellum? Studies in sexually trained male rats found that female bedding or almond smell elicited increased cFos immunoreactive granule cells in the vermis compared to rats exposed to clean Air (Hernandez-Briones, Z. S. et al. *Neurobiol. Learn. Mem.* 146,31–36, 2017).

Minor comments

1. "...lobule IV/V with photostimulation occasionally impeded exploratory (Fig. 2e)". Is this correct? The variance of the data is large.
2. In the methods change "maximus" to "maximum".
3. The data that are shown in Figure 2b do not support the large increase in the coefficient of variation shown in Fig. 2d. Please show a histogram of the interspike interval.

Reviewer #2 (Remarks to the Author):

This is a quite powerful and interesting paper that demonstrates a role of cerebellar vermal areas in social recognition memory (SRM). Chemo- and optogenetic activation of molecular layer interneurons (MLIs) is used to suppress activity in Purkinje cells, and thus the output signal originating from the cerebellar cortex. The observations suggest a cerebellar involvement in social memory retrieval rather than encoding. Also of interest is the finding that MLI manipulation changes neural network coordination from engagement of a corticolimbic network to a stronger activation (inclusion?) of the amygdala. Although this data is largely based on activity-dependent c-FOS labelling, with all its strengths but also limitations, this is very interesting and the limitations are actually nicely discussed as a caveat.

I have two major comments:

First, in this experimental set-up, it obviously is important that increasing activity in MLIs indeed reduces spike firing in Purkinje cells. This is shown for the Chr2 optogenetic activation (Fig. 2b), but it is not shown for the chemogenetic activation (only the CNO-induced increase in MLI firing is shown; Fig. 1c). If there is a reason for this omission, it should be appropriately discussed. If there is no specific reason, it would be better to provide this control. Pointing towards the optogenetic result might not be sufficient as the activation (spatial spread; amplitude etc) is not the same.

Second, the observation and underlying claim that intact Purkinje cell activity engages a corticolimbic network, while disrupting the output from cerebellar cortex causes a shift towards amygdala engagement is so crucial – actually beyond the social memory aspects – that it feels somewhat wrong not to explore this finding further, in a more general manner. This is more a suggestion than a request,

but the authors might want to think about recordings during an amygdala-specific task and then perform behavioral as well as the c-FOS experiments upon MLI activation. Would this lead to a further increase in labelling as well?

Returning to the social context, but with the limbic system / amygdala findings in mind, a previous paper (Simmons et al., *Neuroscience* 462, 2021) has pointed out that social approach in the three-chamber test can be positively motivated (curiosity / pro-social play etc) or negatively motivated (e.g. territorial behavior). It might be of interest to discuss the three-chamber findings presented here in some more depth with this context in mind.

Reviewer #3 (Remarks to the Author):

In this manuscript, Chao and colleagues used a combination of DREADDs and optogenetics to test the role of the cerebellum in social recognition memory (SRM). The authors report several key findings. First, they show that the vermis, particularly the anterior lobules, has a significant impact on SRM. Second, they show that modulating the function of molecular layer interneurons in these lobules is sufficient to drive the behavioral effects. Third, they show that the cerebellum influences retrieval but not encoding of social information. They go on to use computational methods and anatomical tracing to explore a network and circuit model that the cerebellum potentially used to influence SRM. In all, this is a very exciting set of findings from a talented group of collaborators. The use of very clever genetics paired with electrophysiology and behavior provide compelling evidence for cerebellar circuit function in SRM. The figures are very clear and nicely presented. Below I highlight several key points that I hope will help the authors improve the clarity of this already excellent piece of work.

1) In the title, I don't understand what the authors mean by the idea that SRM is "gated by dysregulation" of the cerebellar vermis. Exactly what experiments were conducted to address a gating mechanism?

2) Very minor point, but most authors are now sticking with "cerebellar nuclei" rather than "deep cerebellar nuclei".

3) What exactly do you mean by "...second-order targets virtually in the entire neocortex"? I don't understand.

4) The authors state "While the anterior cerebellum (lobules I-V) is largely involved in motor coordination, the posterior cerebellum (lobules VI-IX) participates in social cognition, working memory, and language by interactions with the prefrontal, parietal, and temporal cortices..." This seems like a very broad over generalization. Especially since the posterior lobules can control spinocerebellar proprioceptive function(motor) and as well as various eye movements and vestibular function(lobules IX and X and flocculus/paraflocculus). This statement should be expanded with additional details. The authors should provide a more comprehensive argument as to the different specific functions and

topographical features throughout the cerebellum. As it stands, the argument of how they chose the specific areas to target is somewhat vague. Also, a more comprehensive description of the vermis versus paravermis versus hemispheres should be provided. The authors somewhat toggle between human anatomy descriptions and relate it to mouse in a general way, which is not wrong but it is confusing considering the experiments that are conducted wish to consider specific circuits and their related topography. In short, please provide a whole lot more detail.

5) Please clarify what you mean by “...Interestingly, dysregulated cerebellar circuits, in which low PC activity at...”. Specifically, how are you defining low Purkinje cell activity? How low, and what in what context and experimental paradigm? Some additional references from the literature could also be helpful. There is mounting evidence about how Purkinje cell activity is altered in different behaviors and disease states, referencing this literature more extensively would be helpful.

6) The authors state that “...techniques, we phenocopied the cerebellar dysfunction observed in the ASD mouse models...” This is a bold statement. First, what aspects of ASD did you phenocopy? What were the criteria for making this comparison? Also, please expand on the ASD phenotypes. As it stands, I am not sure if this was supposed to be part of the main argument or not, but as it stands its currently somewhat peripheral to the main argument.

7) The idea to modulate the MLIs as the primary means of manipulating the circuit is very interesting. However, there is very little rational and justification for how this approach might directly address the main question. Here again, the authors need to greatly expand on the details of MLIs and why altering their activity specifically might be the ideal way of examining cerebellar non-motor function. Later in the manuscript the authors do indeed provide some of these ideas, but those come much too late and additional rationale directly related to overall question needs to be provided upfront.

8) The authors state “The vermis was targeted because of its topographical connection to the limbic system and its consistent association with neuropsychiatric disorders.” What is the evidence that an equivalent pathway exists in mouse? I suppose I am trying to gain a better appreciation for what the authors rationalized about what specific circuits they might modulate. Please expand on this discussion as its currently unclear how the topography will relate to the specific data generated here.

9) For the DREADD experiments, how consistent are the cells that you affect from mouse to mouse? Related, how consistent were the virus injections and how consistent is the infected population by the time the manipulation is carried out? Please provide some anatomy/reporter staining as evidence for the consistency. Currently, only single samples are shown and it is therefore very difficult to appreciate if and whether differential targeting might have an impact on mouse to mouse or regional variability.

10) Related to above, in figure 1b, please provide additional sample tissue sections to gain a better appreciation of what cells were targeted throughout the injected region. Also, additional samples should be provided from different mice and some sort of quantification must be carried out to look at the consistency of injections and expression of the virus.

11) In figure 1b, what are the large cells in the granule cell layer? And how might these impact your interpretation? Were their effects accounted for?

12) Figure 1c shows a relatively modest increase in firing frequency after CNO. What is the ultimate effect on Purkinje cells and/or cerebellar nuclei neurons? It seems surprising that such a modest effect

on what is likely a small subset of MLIs can have such a powerful impact on behavior. In this regard, what is the actual size of the population of MLIs that is affected? Related, what is the effect on the Purkinje cells in terms of how many and what is their location? The authors hint at topography throughout the manuscript and the effect on Purkinje cells is almost certainly going to have a direct effect on topography at several levels. These issues need to be dealt with in full.

13) Figure 2d reports an increase in PC CV when light is ON. However, this is not so clear from the raw trace that is provided. Also, only a limited explanation is provided for how a change in CV is related to the overall phenotype(s) that is reported. Essentially, what does the increased CV mean?

14) The cfos staining in Figure 3 is not so convincing. Perhaps the authors need to show additional adjacent tissue sections and they might also consider showing a larger area as well. As it stands, it is very hard to appreciate any difference from control. Last, some higher power images could also help in this regard.

15) Related to above, what is the identity of the different cfos labeled populations? A little more information about what specific cells increase cfos expression in specific relevant areas would be very helpful.

16) For Figure 6, it's unclear if different animals are shown across the different panels, or which panel goes with which animal. I think it would be useful to show multiple samples, with each case presented in a manner that one could track the projections per animal injected. Also, the example shown in 6b is complicated because it looks like both the fastigial and interposed are labeled. It would be more compelling to see more discrete targeting achieved for each nucleus.

Point-by-point response to reviewer comments on manuscript NCOMMS-22-07490

Overall response: We would like to thank the reviewers for their valuable feedback on our manuscript titled “Social memory deficit caused by dysregulation of the cerebellar vermis”. We are encouraged that they have viewed our work as “novel”, “powerful” and “exciting” with “solid” and “compelling evidence” supporting the role of the cerebellar circuits in social recognition memory. To address their critiques, we have substantially revised the manuscript by adding four sets of new experiments and changing the presentation throughout the text (shown in blue font). Below are our explanations for the revisions. Reviewer comments are italicized, and our response to each point is outlined as follows.

REVIEWER COMMENTS

Reviewer #1 (Remarks to the Author):

The manuscript by Chao and co-workers describes disruption of social recognition memory (SRM) by chemogenetic or optogenetic excitation of cerebellar molecular layer interneurons (MLIs) in lobules VI and VII. In addition, the authors present a cFos measurement of activity in different brain regions involved in the output of the cerebellum and make the interesting finding that chemogenetic excitation of Vermis VI/VII decreases interactions with the corticolimbic system. These results are novel providing a circuit basis for social impairment in autism and other psychiatric disorders and contributes to emerging evidence indicating that the cerebellum is involved in non-motor brain processes. In my opinion this is a solid and interesting manuscript. However, I find problems with data presentation, and lack of a thorough analysis of first order and second order DCN projections in Figure 6.

Author Response: We sincerely appreciate your enthusiasm and your insightful comments about our study. To improve data presentation, we have included data points in all plots (e.g., Fig. 1-3). For larger sample sizes (>10), we have opted for box-and-whisker plots (Fig. 4d-4h), in compliance with the journal format. To conduct a thorough analysis of the anatomical data, we have (1) shown representative images from all 24 brain regions that constitute the network for social recognition memory (SRM) (Fig. 6, 7, Supplementary Fig. 5); (2) quantified the abundance of cerebellar projections in these regions (Supplementary Fig. 5i); and (3) summarized the findings in a schematic model (Supplementary Fig. 7). Additionally, to address related comments from other reviewers, we have incorporated retrograde tracing of the amygdalar input into the anterograde tracing of the cerebellar output (Fig. 8).

Major comments:

1. The analysis of DCN cerebellar projections in Figure 6 is disappointing. Please provide a more thorough analysis and a summary diagram. How do these findings relate to published findings by other investigators?

Author Response: Thank you for pointing this out. To strengthen the anatomical study and respond to a similar concern from other reviewers, we have refined the neural tracing

experiments by precisely targeting the fastigial nuclei (FN). In the early submission, the virus-based tracing spread from the FN to the interposed nuclei. Knowing that Purkinje cells (PCs) in the vermis mainly project to the FN¹, the new results represent a more accurate structural map of the vermal output (Fig. 6, 7, Supplementary Fig. 5).

To conduct a thorough analysis of the FN output, we have examined the confocal images taken from all 24 brain regions in the SRM network (Fig. 6, 7, Supplementary Fig. 5). The brain sections were co-labeled with DAPI (a marker for cell nuclei) to better identify neuronal clusters in each region. To quantify the abundance of FN projections, we have normalized the intensity of tdTomato labeling to the background in each region of interest (40 μ m-thick serial sections for each region) and summarized the results for 24 regions from 3 individual mice (Supplementary Fig. 5i). The original data can be found in the Source Data file.

To address a further comment from Reviewer #2, we have incorporated retrograde tracing of the amygdala input into the anterograde tracing of the FN output (Fig. 8, Supplementary Fig. 6). Despite a lack of direct connection to the FN, the amygdala is highly integrated in the FN-cortex circuits. A schematic diagram to summarize the findings is shown in Supplementary Fig. 7.

Overall, our data is consistent with anatomical studies in the literature. For example, we have specifically compared our data with a recent study on the FN output in the mouse brain using a similar approach². As stated in the Results (Page 12-13), the only discrepancy is that we observed sparse disynaptic projections from the FN to the infralimbic cortex (IL), but this was not reported in the previous study². We have further explained the variation, which is “possibly due to differences in injection sites or floxed mice (Ai9 vs Ai14)”. Our experiments are not designed to replicate previous anatomical studies but rather provide a structural basis for the cerebellar engagement in the SRM network. For this purpose, the combined neuronal tracing of the BLA (basolateral amygdala) input and the FN output has offered compelling evidence for a specific role of the cerebellum in emotion-based memory (Fig. 1-2).

2. The presentation of the data is uneven making it difficult for the reader to evaluate the conclusions. First, some of the figures show individual data points (e.g., Fig. 1e, f, g) while other figures do not (e.g., Fig. 3b, c). Please show individual points in all figures. Second, the bar graphs show the standard error of the mean that does not provide information on whether there are statistical differences. A better measure is to show the bootstrapped 95% confidence interval (Hasley et al 2015 <https://doi.org/10.1038/nmeth.3288>) or the standard deviation.

Author Response: Thank you for the suggestions. We have included individual data points in all plots (Fig. 1-3, Supplementary Fig. 2, 3). For larger sample sizes (>10), box-and-whisker plots are presented as an alternative (Fig. 4d-h), in compliance with the journal format. The original data used for the graphs can be found in the Source Data file.

We agree with the reviewer that standard deviation is a better measure of variability within a single sample. We have calculated standard deviation (SD) and standard error of mean (SEM) for all the data and described them in the Source Data file. As most publications in the journal use the SEM for plotting bar graphs, we have kept this common practice. Statistical differences

in all figures are indicated with an asterisk (*) for “significant” or *ns* for “not significant”. The details on statistical methods and results (e.g., *p* values) are listed in Supplementary Table 1.

3. For Figure 4 please provide examples of the graphs showing the correlation of cFos expression between different brain areas that was used to calculate the Pearson correlation coefficients. Please state the number of mice per group. Is this analysis appropriately powered? Did the cFos measurements include cFos measurement in DCN and the cerebellum? Studies in sexually trained male rats found that female bedding or almond smell elicited increased cFos immunoreactive granule cells in the vermis compared to rats exposed to clean Air (Hernandez-Briones, Z. S. et al. *Neurobiol. Learn. Mem.* 146,31–36, 2017).

Author Response: Thank you for raising these points. For Fig. 4, the data came from c-Fos immunostaining shown, for instance, in Fig. 3. The c-Fos was measured by counting the number of c-Fos-positive cells in 2-4 serial sections for each region. The Pearson correlation coefficient (*r*) was calculated for each pair of brain regions across the subjects in each group (*n*=6 mice/group). For example, in the control group, c-Fos counts for the anterior cingulate cortex [rostral] (Acg) were 79.25, 65.73, 86, 153.33, 86.33, 88.8; and for the prelimbic cortex (PL) were 89.17, 113.6, 221.17, 168.67, 220.33, 133.33. Each value was from an individual mouse and represented the average number of c-Fos-positive cells from the serial sections for each region. We correlated the c-Fos counts in the Acg with those in the PL and obtained a Pearson correlation of 0.242. Simply put, each Pearson correlation (presented as a single square in Fig. 4a) contained a dataset from 24-48 brain sections. The procedure was then repeated across 24 regions, e.g., Acg x Acg, Acg x PL, Acg x IL, ..., to gain the correlation matrix. Given the immense datasets, it would not benefit readers to show individual examples. The heat maps that indicate the strength of the correlation between different brain regions (Fig. 4a) provide a clearer view of the discrete pattern of neural networks for each group. To ensure data transparency, we have provided the original data and calculations in the Source data file.

As stated in the Fig. 4 legend, there were 6 mice per group (18 in total for 3 groups). In each animal, we examined 24 brain regions by averaging c-Fos counts from 2-4 sections for each region. The sample size for each group ranged from 288 to 576, when all 24 regions were considered to generate the mean *r* (Fig. 4d). Even though the sample sizes were smaller when the *r* were calculated for individual regions (Fig. 4e-h), they were adequate for statistical analysis. One-way ANOVA was used to analyze the differences across all groups. After finding a significant difference, post hoc Fisher’s LSD was used to determine which specific groups differed from one another. The statistical details are included in Supplementary Table 1.

The cerebellar cortex and cerebellar nuclei were not included in the c-Fos measurements because they were taken for examining the distribution of fluorescent proteins as indicators for the viral targeting (e.g., Fig. 1, 6, Supplementary Fig. 1). This information is provided in the Methods section (Page 25). Also, we have discussed the interesting study that showed the cerebellum could be reactive to external stimuli such as odorants (Page 18).

Minor comments:

1. "...lobule IV/V with photostimulation occasionally impeded exploratory (Fig. 2e)". Is this correct? The variance of the data is large.

Author Response: This is correct. Photostimulation of molecular layer interneurons in lobule IV/V led to a reduction in total exploration time in the social recognition test, as compared to other groups (new Fig. 2g). We have explained this observation as such that it may "reflect deficits in motivation, attention and/or sensorimotor integration associated with interference of the anterior cerebellum" (Results, Page 8, Line 14-17), based on a previous study where we showed neurotoxic lesion of the anterior vermis impaired animals' exploratory activity³. However, it is worth mentioning that the photostimulation did not affect the total exploration time in other tests (new Fig. 2h, i). More importantly, it did not affect social preference or social memory, despite that the exploratory behavior was sometimes impeded (new Fig. 2g).

We agree with the reviewer that the variance in mouse behavioral data can be large. Generally, this is attributed to genetic and environmental factors. To limit genetic variations, we used littermate controls with the same background to test behaviors after chemo- and optogenetic manipulations (Results and Methods, Page 6, 8, 20). To control environmental variations, we conducted behavioral tests under the same conditions for all groups. For example, all tests were done in a quiet room (<40 dB) under dim light (~50 lx) during 10:00-16:00 (Methods, Page 22). All animals were allowed to habituate to the apparatus before testing (e.g., in the three-chamber social test; Methods section, Page 23). To address individual variations, we included a good number (8-10) of animals in each group. The sample size was based on previous studies using similar experimental protocols. One-way or two-way mixed ANOVA was used to analyze the differences across all groups. Unpaired or paired *t*-tests were applied when main effects were found (Methods, Page 27). Collectively, we have taken measures to mitigate the variability issue related to the nature of these assays. The raw data and statistical analysis were provided in the Source Data file and Supplementary Table 1, respectively, to allow transparency and reproducibility of our results.

2. In the methods change "maximus" to "maximum".

Author Response: Thank you! This is corrected (Page 21, 22).

3. The data that are shown in Figure 2b do not support the large increase in the coefficient of variation shown in Fig. 2d. Please show a histogram of the inter-spike interval.

Author Response: Thank you for pointing this out. We have improved the presentation by selecting more representative traces with a longer time frame to show the overall increase in the coefficient variation of inter-AP intervals in PCs following the photostimulation (Fig. 2b). By plotting a histogram with a Gaussian fit of the inter-AP intervals, we found that the stimulation created two distinct peaks centering around 14.5 ms and 54.2 ms, in contrast to the unimodal pattern with a peak around 17.5 ms without stimulation (Fig. 2f). Based on the mean values of

the Gaussian distribution, the 1st peak likely represents spontaneous PC activity (higher frequency) whereas the 2nd peak may correlate to the stimulus pattern (lower frequency). This finding is summarized in the Results (Page 8). Together, your feedback has helped us gain new insights into how the manipulation affects PC activity and ultimately social behavior.

Reviewer #2 (Remarks to the Author):

This is a quite powerful and interesting paper that demonstrates a role of cerebellar vermal areas in social recognition memory (SRM). Chemo- and optogenetic activation of molecular layer interneurons (MLIs) is used to suppress activity in Purkinje cells, and thus the output signal originating from the cerebellar cortex. The observations suggest a cerebellar involvement in social memory retrieval rather than encoding. Also of interest is the finding that MLI manipulation changes neural network coordination from engagement of a corticolimbic network to a stronger activation (inclusion?) of the amygdala. Although this data is largely based on activity-dependent c-FOS labelling, with all its strengths but also limitations, this is very interesting, and the limitations are actually nicely discussed as a caveat.

Author Response: We sincerely appreciate your enthusiasm and your insightful comments about our study. To address your critiques, we have done new experiments to show (1) how chemogenetic activation of molecular layer interneurons (MLIs) inhibits Purkinje cell (PC) firing activity with electrophysiology (Fig. 1e, f); and (2) how the amygdala, particularly basolateral amygdala (BLA), is integrated in the network for social recognition memory (SRM) through retrograde tracing of the amygdalar input and anterograde tracing of the cerebellar output (Fig. 8, Supplementary Fig. 5, 6). It is important to note that the anatomical connections between the cerebellar vermis and BLA are not linear but rather integrative (Supplementary Fig. 7). Future studies are needed to explore their functional connectivity in a task-specific manner. Overall, our work represents the first step in identifying the circuit basis for cerebellar engagement in SRM.

I have two major comments:

First, in this experimental set-up, it obviously is important that increasing activity in MLIs indeed reduces spike firing in Purkinje cells. This is shown for the Chr2 optogenetic activation (Fig. 2b), but it is not shown for the chemogenetic activation (only the CNO-induced increase in MLI firing is shown; Fig. 1c). If there is a reason for this omission, it should be appropriately discussed. If there is no specific reason, it would be better to provide this control. Pointing towards the optogenetic result might not be sufficient as the activation (spatial spread; amplitude etc.) is not the same.

Author Response: Thank you for pointing this out. We have done patch-clamp recordings from PCs in cerebellar slices taken from c-kit^{ires-Cre} mice that were transduced with AAV8-hSyn-DIO-hM3Dq-mCherry. Activation of the hM3Dq with CNO boosted the activity of basket cells (BCs), a subtype of MLIs that provide major inhibition on PCs (Fig. 1c, d). Due to feedforward inhibition, CNO decreased the firing rate and increased the firing variation of PCs that were connected to hM3Dq-expressing BCs. In contrast, CNO did not affect PCs that were innervated by non-

transduced BCs (Fig. 1e, f). Using the same mice, we also labeled PCs with an anti-Calbindin antibody to examine the BC-PC microcircuits and confirmed that a BC can innervate several adjacent PC somas (Supplementary Fig. 1). These results demonstrate the effectiveness and specificity of the chemogenetic manipulation, as shown in a previous study ⁴.

Second, the observation and underlying claim that intact Purkinje cell activity engages a corticolimbic network, while disrupting the output from cerebellar cortex causes a shift towards amygdala engagement is so crucial – actually beyond the social memory aspects – that it feels somewhat wrong not to explore this finding further, in a more general manner. This is more a suggestion than a request, but the authors might want to think about recordings during an amygdala-specific task and then perform behavioral as well as the c-FOS experiments upon MLI activation. Would this lead to a further increase in labelling as well?

Returning to the social context, but with the limbic system / amygdala findings in mind, a previous paper (Simmons et al., Neuroscience 462, 2021) has pointed out that social approach in the three-chamber test can be positively motivated (curiosity / pro-social play etc.) or negatively motivated (e.g., territorial behavior). It might be of interest to discuss the three-chamber findings presented here in some more depth with this context in mind.

Author Response: Thank you for the suggestions. We agree that the amygdala, particularly BLA, is an intriguing hub in the vermis involved SRM network (Fig. 3-5). To explore further, we have done new experiments and added a section in the Results (Page 13-14). We infused AAV1-hSyn-Cre in the fastigial nucleus (FN) and AAVrg-hSyn-EGFP in the BLA of Ai9 mice (Fig. 8). As AAV1-hSyn-Cre has an anterograde transsynaptic property ⁵, it labeled the FN and its downstream (first- and second order target) neurons with tdTomato (Fig. 6, 7). Being a retrograde virus ⁶, AAVrg-hSyn-EGFP labeled the BLA and its upstream neurons with EGFP (Fig. 8). Using these two fluorescent signals, we probed the anatomical relationship between the FN and BLA. In line with the literature, the BLA was connected to a wide range of brain regions (Supplementary Fig. 6). For example, in the cortices that were identified as the second-order FN targets (Fig. 7), we noticed EGFP-expressing neurons, but they rarely made close contact with axons expressing tdTomato (Fig. 8h, i). The separate signals were observed in all brain regions, suggesting that the FN does not connect with the BLA through hierarchical neurotransmission. The impact of the vermis on amygdalar activity and connectivity (Fig. 3-5) is likely through their interactions with other cortical or subcortical regions. Yet, the coexistence of the FN outputs and BLA inputs in these regions indicates a role of the BLA in integrating sensory and emotional information into the FN-neocortex circuits. A schematic diagram to summarize the findings is shown in Supplementary Fig. 7. Given the complexity of the integrated network, we share the feeling with the reviewer that understanding of the functional connectivity between the cerebellum and amygdala will require future work using in vivo electrophysiology or Ca²⁺ imaging to monitor real-time neuronal activity in a task-dependent manner (Discussion, Page 16-17). Nevertheless, our study represents the first step in identifying the circuit basis for cerebellar involvement in SRM and opens new avenues for future research.

Regarding the confounding factors in assessment of social behavior, we have cited the review by Simmons et al (Introduction, Page 5; Discussion, Page 14). This is indeed an important point because it has been shown that perturbing the vermis can lead to changes in exploration and aggression^{3,7,8}. Our strategy of exciting MLIs instead of directly inhibiting PCs to attenuate, rather than silence, the output of the cerebellar cortex may have helped control these factors (Introduction, Page 5). In fact, this manipulation has specifically impaired SRM without affecting sociability, anxiety, locomotion, novelty seeking, or object-based recognition memory (Fig. 1, 2). Regarding the positive motivations for social interactions, we have discussed the possibility that “the cerebellum may promote SRM through its connections to the reward pathway, which significantly overlaps with the social brain network” (Discussion, Page 16).

Reviewer #3 (Remarks to the Author):

In this manuscript, Chao and colleagues used a combination of DREADDs and optogenetics to test the role of the cerebellum in social recognition memory (SRM). The authors report several key findings. First, they show that the vermis, particularly the anterior lobules, has a significant impact on SRM. Second, they show that modulating the function of molecular layer interneurons in these lobules is sufficient to drive the behavioral effects. Third, they show that the cerebellum influences retrieval but not encoding of social information. They go on to use computational methods and anatomical tracing to explore a network and circuit model that the cerebellum potentially used to influence SRM. In all, this is a very exciting set of findings from a talented group of collaborators. The use of very clever genetics paired with electrophysiology and behavior provide compelling evidence for cerebellar circuit function in SRM. The figures are very clear and nicely presented. Below I highlight several key points that I hope will help the authors improve the clarity of this already excellent piece of work.

Author Response: We sincerely appreciate your enthusiasm and your insightful comments about our study. Your feedback has been instrumental in helping us strengthen the rationale for targeting molecular layer interneurons (MLIs) in the vermis, enhance the clarity of data presentation, and improve the interpretation of our results. Due to the constraints on word count and reference numbers, at times we couldn't extensively discuss the topography of the whole cerebellum or the physiopathology of Purkinje cells (PCs). However, we have prioritized our effort to address several closely related issues by adding four sets of new experiments and making revisions throughout the text.

1) In the title, I don't understand what the authors mean by the idea that SRM is “gated by dysregulation” of the cerebellar vermis. Exactly what experiments were conducted to address a gating mechanism?

Author Response: Thank you for pointing this out. We have changed the title to “...caused by dysregulation...” to better summarize the main finding that perturbation of the cerebellar vermis leads to a social memory deficit.

2) *Very minor point, but most authors are now sticking with “cerebellar nuclei” rather than “deep cerebellar nuclei”.*

Author Response: Thank you! This is corrected throughout the text (e.g., Introduction, Page 3).

3) *What exactly do you mean by “...second-order targets virtually in the entire neocortex”? I don't understand.*

Author Response: We apologize for the confusion. We have rephrased the text, which reads “Neurons in the CN directly project to subcortical regions (first-order targets) such as the ventral thalamus (vTH). Via the first-order neurons, the CN connects to second-order targets virtually in the entire neocortex” (Introduction, Page 3). The first-order targets refer to direct synaptic connections between two regions, e.g., the cerebellar nuclei (CN)-to-thalamus connections. The second-order (or disynaptic) targets are indirect and involve an intermediate structure, e.g., the CN-to-neocortex connections through the thalamus. The organization of these connections can be revealed using neural tracing techniques. In this study, we injected AAV1-hSyn-Cre into the fastigial nucleus of floxed tdTomato (Ai9) mice to label fastigial neurons via Cre-LoxP recombination (Fig. 6a, b). As this virus has a transsynaptic property ⁵, it was able to further transduce neurons in the first-order target areas as indicated by the presence of labeled somas (Fig. 6c). The second-order target areas were identified by the presence of distal axons and axonal terminals of the labeled first-order neurons (Fig. 7). The experimental details are described in the Methods (Page 20, 26).

The extensive cerebellum-neocortex connections are well recognized ⁹. For instance, recent studies in the mouse brain using a similar approach (viral neuronal tracing) have shown that the CN, even the fastigial nucleus alone, disynaptically project to a wide range of cortical regions including the frontal cortex, sensory/motor cortex, and association cortex ^{2,10}.

4) *The authors state “While the anterior cerebellum (lobules I-V) is largely involved in motor coordination, the posterior cerebellum (lobules VI-IX) participates in social cognition, working memory, and language by interactions with the prefrontal, parietal, and temporal cortices...” This seems like a very broad over generalization. Especially since the posterior lobules can control spinocerebellar proprioceptive function(motor) and as well as various eye movements and vestibular function (lobules IX and X and flocculus/paraflocculus). This statement should be expanded with additional details. The authors should provide a more comprehensive argument as to the different specific functions and topographical features throughout the cerebellum. As it stands, the argument of how they chose the specific areas to target is somewhat vague. Also, a more comprehensive description of the vermis versus paravermis versus hemispheres should be provided. The authors somewhat toggle between human anatomy descriptions and relate it to mouse in a general way, which is not wrong, but it is confusing considering the experiments that are conducted wish to consider specific circuits and their related topography. In short, please provide a whole lot more detail.*

Author Response: Thank you for the suggestions. To revise the overgeneralization, we have elaborated the topographical organization of the cerebello-cortical circuits with clarification on the motor functions of posterior lobules IX and X. The new text states: “Functional magnetic resonance imaging (fMRI) of the human brain has revealed that lobules I-V and VIII are involved in sensorimotor tasks via interactions with the somatosensory and motor cortices; lobules VI-VII in social cognition, working memory, and language via interactions with the prefrontal, temporal and parietal cortices; lobules IX and X (flocculus) in eye movement and balance via interactions with the vestibular system”. Further, we have emphasized that “these functions are generalized because motor and non-motor dichotomy of certain lobules can occur” (Introduction, Page 3).

To strengthen the argument for manipulating the vermis, we have provided more details on the anatomy and functions of the vermis (Introduction, Page 3-4). As described in a book chapter ¹, a characteristic function of the vermis is to regulate emotion via interactions with the limbic system. Schmahmann and Sherman first reported patients with damage to the posterior vermis displayed cognitive and affective symptoms ¹¹ (Discussion, Page 16). Since then, many studies have shown that the vermis is activated during emotional processing. A new finding suggests that the human cerebellum, mainly within the vermis, is part of the corticocerebellar circuits for enhancing emotional memory ¹². Reduced volume or activity of the vermis is consistently associated with neuropsychiatric disorders ¹, e.g., autism spectrum disorder (ASD), which is characterized by social interaction deficits. Moreover, animal studies using neural tracing methods have demonstrated that the vermis is extensively connected with cognitive, affective, and motor forebrain circuits, including the corticolimbic system such as the medial prefrontal cortex (mPFC) and anterior cingulate cortex (ACC) ^{2,10}. This provides a structural base for the behavioral alterations in aggression, spatial memory, and social association by vermal perturbation ^{3,7,8} (Introduction, Page 4). Collectively, we have presented additional justification for investigating the role of the vermis in social behavior.

While adding the justification in the Introduction, Results and Discussion sections, we have now clearly indicated whether the evidence is from human or animal studies.

Given the strict limits on words and references, we could not *comprehensively* describe the differences between the vermis, paravermis and hemispheres. We sincerely apologize for this. However, we have taken steps to prioritize several related issues. To better distinguish the vermis from paravermis and to address comment #16, we have redone the neural tracing study by precisely targeting the FN. In the early submission, the virus-based tracing spread from the FN to the interposed nuclei. As PCs in the vermis mainly project to the FN ¹, our new results represent a more accurate structural map of the vermal output (Fig. 6, 7, Supplementary Fig. 5). To know how the vermis interacts with the amygdala, a key component in the limbic circuits, and to address a relevant comment from other reviewers, we have incorporated retrograde tracing of the amygdalar input into the anterograde tracing of the cerebellar output (Fig. 8). A schematic diagram (Supplementary Fig. 7) illustrates that the FN does not connect with the amygdala (specifically basolateral amygdala, BLA) through hierarchical neurotransmission. Instead, the BLA is intertwined in the FN-neocortex network, together to support the emotion-based SRM.

5) Please clarify what you mean by “...Interestingly, dysregulated cerebellar circuits, in which low PC activity at...”. Specifically, how are you defining low Purkinje cell activity? How low, and what in what context and experimental paradigm? Some additional references from the literature could also be helpful. There is mounting evidence about how Purkinje cell activity is altered in different behaviors and disease states, referencing this literature more extensively would be helpful.

Author Response: We agree with the reviewer that there is a large body of literature showing PC activity is altered in different behaviors and disease states. We couldn't quote individual studies due to the limitation on the number of references. To overcome it, we have cited a recent review on this topic¹³. Here, the PC activity refers to spontaneous action potentials (APs) elicited from PCs (Fig. 1, 2), encoding the output information of the cerebellar cortex. Reduced PC activity leads to motor and non-motor dysfunctions in neurodegenerative (e.g., ataxia) and neurodevelopmental (e.g., ASD) disorders, although it is difficult to pinpoint whether the extent of PC firing deficits, the lobule-specific targeting of genetic mutations, or the conditions of other brain regions result in certain behavior phenotypes in these disorders (Discussion, Page 15).

To examine how low the PC activity was in ASD mouse models¹⁴⁻¹⁷, we checked the original reports and quantified the frequencies of APs recorded from PCs in cerebellar slices in the cell-attached mode using the patch-clamp technique. On average, the PC firing frequencies decreased by 35-55%, compared to the wildtype controls. As the experimental conditions in these studies were variable (e.g., recording temperature and mouse genetic background), we couldn't simply correlate the level of PC firing impairments to the severity of ASD-like behaviors. However, it is noteworthy that restoring PC activity is to date the most effective avenue to rescue the social deficits in these ASD models¹³. This information is added in the Discussion (Page 15).

6) The authors state that “...techniques, we phenocopied the cerebellar dysfunction observed in the ASD mouse models...” This is a bold statement. First, what aspects of ASD did you phenocopy? What were the criteria for making this comparison? Also, please expand on the ASD phenotypes. As it stands, I am not sure if this was supposed to be part of the main argument or not, but as it stands its currently somewhat peripheral to the main argument.

Author Response: The reviewer is correct that cerebellar abnormalities in ASD patients and animal models are multifaceted, including PC loss and impaired PC activity and plasticity that underlie a spectrum of behavioral deviations such as deficits in motor learning and social interaction (Discussion, Page 15). This statement is supported with two influential reviews^{18,19}. Among the cerebellar phenotypes, reduced PC activity is consistently observed in the ASD mouse models¹⁴⁻¹⁷. More importantly, restoring PC activity is shown to be the most effective approach to rescue the social deficits in these mice¹³. Thereby, we have decided to target this specific phenotype to probe the role of the cerebellum in social behavior.

We have also explained that PC activity is strongly controlled by feedforward inhibition from MLIs (Introduction, Page 3). The effects of MLI inhibition on PCs are demonstrated in Figures 1 and 2, where MLIs were activated by the chemo- or optogenetic manipulation. Using animal

models for ASD, we and others have revealed that excessive GABA release from MLIs contributes to the low PC activity¹⁵⁻¹⁷. Evidence from postmortem analysis of ASD patients also shows an increase of GAD67 mRNA in MLIs, implying dysregulated inhibition on PCs²⁰. To avoid the ambiguity of “phenocopy”, we now clearly state that “we mimicked the dysregulated MLI-PC inhibition observed in ASD patients and mouse models by selectively increasing the excitability of MLIs to reduce PC firing...” (Introduction, Page 5). We hope these revisions have strengthened our argument for manipulating the MLI-PC pathway. More rationale is provided in our response to the next comment.

7) The idea to modulate the MLIs as the primary means of manipulating the circuit is very interesting. However, there is very little rationale and justification for how this approach might directly address the main question. Here again, the authors need to greatly expand on the details of MLIs and why altering their activity specifically might be the ideal way of examining cerebellar non-motor function. Later in the manuscript the authors do indeed provide some of these ideas, but those come much too late and additional rationale directly related to overall question needs to be provided upfront.

Author Response: We thank the reviewer for considering our approach is interesting and for offering the suggestions. To better justify for manipulating MLIs, we have:

- A. moved up the paragraph about how MLIs define motor and non-motor functions of the cerebellum to the Introduction section (Page 4).
- B. elaborated the role of MLIs in the pathogenesis of ASD, a disorder characterized by social interaction deficits (Page 4). The current text includes evidence from postmortem analysis of ASD patients showing increased GAD67 mRNA in MLIs that implies dysregulated inhibition on PCs²⁰, and evidence from ASD animal models where MLI over-inhibition leads to reduced PC activity and social impairments. Notably, we have shown that controlling GABA release by targeting Kv1.2 channels in the MLI nerve terminals effectively alleviates social deficits in various ASD mouse lines^{16,17}.
- C. explained a technical advantage of exciting MLIs over inhibiting PCs directly. This strategy to attenuate, rather than silence, output activity of the cerebellar cortex has helped limit confounding factors that could affect assessment of social behaviors²¹, such as changes in movement, territorial aggression, or spatial memory when the cerebellar vermis is perturbed^{3,7,8}. This is emphasized in the Introduction (Page 5) and Discussion (Page 14).

8) The authors state “The vermis was targeted because of its topographical connection to the limbic system and its consistent association with neuropsychiatric disorders.” What is the evidence that an equivalent pathway exists in mouse? I suppose I am trying to gain a better appreciation for what the authors rationalized about what specific circuits they might modulate. Please expand on this discussion as its currently unclear how the topography will relate to the specific data generated here.

Author Response: Thank you for pointing this out. A relevant comment (#4) was raised earlier. To address both, we have strengthened the rationale for targeting the vermis by providing more details on the anatomy and functions of the vermis (Introduction, Page 3-4). As described in the book chapter ¹, the vermis and its projections in the FN are termed as the “limbic cerebellum” due to their interactions with the limbic circuits in regulating affect and cognition. This concept originated from early studies in animals. For example, Heath et al showed that stimulating the vermis and the FN generated responses in the limbic regions including the hippocampus and amygdala in cats and rats ²². In contrast, stimulating the cerebellar hemispheres and dentate nuclei yielded no response in these areas, although the anatomical connections were unclear at that time. Recent development in neuronal tracing techniques has provided the evidence, as stated in the text, “... the FN directly projects to the thalamus and hypothalamus ^{2,10,23,24}, although its links to the hippocampus and amygdala may be indirect ². From the thalamus, the FN connects to the limbic cortex including the mPFC and ACC ^{2,10}.” (Introduction, Page 4). These findings are corroborated in our analysis, which showed monosynaptic links of the FN to the subcortical limbic areas (i.e., thalamus and hypothalamus; Fig. 6) and disynaptic links to the cortical limbic areas (e.g., mPFC and ACC; Fig. 7). While describing these circuits in the Results (Page 11-13), we have also briefly explained their functional implications. In the Discussion (Page 16), we highlight the importance of the vermal output to the limbic system for mediating the cognitive and affective processes in SRM.

Specifically related to the main finding in this study, our graph theoretical analysis of the SRM network pinpoints the amygdala (BLA in particular) as a central hub after vermal manipulation (Fig. 5). This may underlie the specific role of the vermis in emotion-involved SRM but not in object-based memory (Fig. 1, 2), despite a lack of mono- or disynaptic projections from the FN to the amygdala ² (see also Fig. 7d). Inspired by the reviewer’s comment on the structural base for SRM, we have explored the anatomical relationship between the FN and the BLA by infusing AAV1-hSyn-Cre in the FN and AAVrg-hSyn-EGFP in the BLA of Ai9 mice (Fig. 8). As AAV1-hSyn-Cre has an anterograde transsynaptic property ⁵, it labeled the FN and its downstream (first- and second order target) neurons with tdTomato (Fig. 6, 7). Being a retrograde virus ⁶, AAVrg-hSyn-EGFP labeled the BLA and its upstream neurons with EGFP (Fig. 8). In line with the literature, the BLA was connected to a wide range of brain regions (Supplementary Fig. 6). For example, in the cortices that were identified as the second-order FN targets (Fig. 7), we noticed EGFP-expressing neurons, but they rarely made close contact with axons expressing tdTomato (Fig. 8h, i). The separate signals were observed in all brain regions, suggesting that the FN does not connect with the BLA through hierarchical neurotransmission. The impact of the vermis on amygdalar activity and connectivity (Fig. 3-5) is likely through their interactions with other cortical and subcortical regions. Yet, the coexistence of the FN outputs and BLA inputs in these regions indicates a role of the BLA in integrating sensory and emotional information into the FN-neocortex circuits. A schematic diagram to summarize the findings is shown in Supplementary Fig. 7. We interpret the data as such that “In general, direct synaptic contacts convey more information than indirect ones. However, the indirect FN-BLA connections may be crucial for processing complex information to modulate the effects of emotion on memory ^{25,26}. The fact that cerebellar perturbation can alter these distal regions supports our view that the cerebellum is an integral part of the cerebello-cortical network for SRM.” (Discussion, Page 16-17).

Although we have focused on the vermis based on the literature ¹ (see also our response to comment #4), further inquiries are needed to understand whether and how other cerebellar subregions are involved in SRM. Our study represents the first step in identifying the circuit basis for cerebellar engagement in SRM and opens new paths for future research.

9) For the DREADD experiments, how consistent are the cells that you affect from mouse to mouse? Related, how consistent were the virus injections and how consistent is the infected population by the time the manipulation is carried out? Please provide some anatomy/reporter staining as evidence for the consistency. Currently, only single samples are shown, and it is therefore very difficult to appreciate if and whether differential targeting might have an impact on mouse to mouse or regional variability.

Author Response: Thank you for raising this important issue. We have done new experiments and further analysis to evaluate the expression of the DREADD (hM3Dq) receptor. To show the individual variability, we included more cerebellar sections (6 in total) taken from different mice (Fig. 1b and Supplementary Fig. 1). Despite a degree of variation, robust yet localized hM3Dq expression (marked with mCherry) was found in lobule IV/V or VI/VII. Moreover, we examined the variability within the individual cerebellum, the hM3Dq was mostly present in midsagittal sections and diminished in sections >500 μm away from the midline (Supplementary Fig. 1), verifying the exact targeting of the central vermis. By normalizing the area containing mCherry to the total area of lobule IV/V or VI/VII in the midsagittal sections, we found $29.2 \pm 2.7\%$ of lobule IV/V and $35.8 \pm 3.0\%$ of lobule VI/VII ($n=6$ mice/group) were affected. The spatial distribution of the hM3Dq in the lobules was indicated in Supplementary Fig. 1b. This information is stated in the Results section (Page 5).

We fixed and imaged the brain sections right after behavioral testing. So, the images embodied the expression of the hM3Dq receptor by the time when it was activated during behavioral tests. We did not find a correlation between the size of transduced areas and the level of behavioral changes (data not shown). Thus, it is unlikely that the transduction variation among different individuals had an impact on the overall behavioral results.

We co-labeled PCs, the output neurons in the cerebellar cortex, with an anti-Calbindin antibody in these sections. As basket cells (BCs), a subtype of MLIs, provide major inhibition on PCs (Fig. 1c, d), we imaged the BC-PC microcircuits and confirmed that a BC innervated several adjacent PC somas (Supplementary Fig. 1c). These results demonstrate the effectiveness and specificity of the chemogenetic manipulation, as shown in a previous report ⁴.

10) Related to above, in figure 1b, please provide additional sample tissue sections to gain a better appreciation of what cells were targeted throughout the injected region. Also, additional samples should be provided from different mice and some sort of quantification must be carried out to look at the consistency of injections and expression of the virus.

Author Response: As described in our previous response, we have included more cerebellar sections taken from different mice in Fig. 1b and Supplementary Fig. 1. Despite a degree of

variation, the hM3Dq (marked with mCherry) was found in lobule IV/V or VI/VII. We assessed the within-subject and between-subject variations in the hM3Dq expression. As the hM3Dq was mostly present in the midsagittal sections (Supplementary Fig. 1a), we quantified its expression by normalizing the area containing mCherry to the total area of lobule IV/V or VI/VII in these sections. We found $29.2 \pm 2.7\%$ of lobule IV/V and $35.8 \pm 3.0\%$ of lobule VI/VII ($n=6$ mice/group) were affected (Results, Page 5). The location of the hM3Dq in the lobules was illustrated in Supplementary Fig. 1b.

To know what cells were targeted, we took a higher-resolution confocal image and showed that the hM3Dq receptor was predominantly expressed in MLIs (Fig. 1b), consistent with an early report⁴. This was further confirmed by co-labeling PCs with an anti-Calbindin antibody to exhibit the BC-PC synaptic connections (Supplementary Fig. 1c).

11) In figure 1b, what are the large cells in the granule cell layer? And how might these impact your interpretation? Were their effects accounted for?

Author Response: We thank the reviewer for pointing this out. As mentioned in the original report on $c\text{-kit}^{\text{IRES-Cre}}$ mice⁴, limited Cre activity could appear in Golgi and glial cells. So, the large cells in the granular layer were likely Golgi cells, although the conditional transduction of MLIs was predominant (Fig. 1b). This caveat is explained in the Results (Page 5). However, we don't think it has compromised the results for two reasons:

- A. As shown in the new Fig. 1c-1f, activation of the hM3Dq receptor with clozapine-N-oxide (CNO) increased the activity of hM3Dq-expressing BCs and suppressed their downstream PCs. CNO had no effect on non-transduced BCs and their connected PCs. This displays the robust effect of feedforward inhibition from MLIs to PCs.
- B. The behavioral outcomes from activation of the hM3Dq in the $c\text{-kit}^{\text{IRES-Cre}}$ mice might have involved a small number of Golgi and glial cells in addition to MLIs. But photostimulation of the nNOS-ChR2 mice only excited MLIs (no contamination from other cell types) produced the same SRM deficit as the chemogenetic manipulation (Fig. 2), arguing for the significant contribution of MLIs to SRM. This is added in the discussion (Page 15-16).

12) Figure 1c shows a relatively modest increase in firing frequency after CNO. What is the ultimate effect on Purkinje cells and/or cerebellar nuclei neurons? It seems surprising that such a modest effect on what is likely a small subset of MLIs can have such a powerful impact on behavior. In this regard, what is the actual size of the population of MLIs that is affected? Related, what is the effect on the Purkinje cells in terms of how many and what is their location? The authors hint at topography throughout the manuscript and the effect on Purkinje cells is almost certainly going to have a direct effect on topography at several levels. These issues need to be dealt with in full.

Author Response: We thank the reviewer for the thoughtful comment. Reviewer #2 also raised a similar concern about how chemogenetic manipulation of MLIs affected PC activity. To this end, we have done patch-clamp recordings from PCs in cerebellar slices taken from $c\text{-kit}^{\text{IRES-Cre}}$

mice transduced with AAV8-hSyn-DIO-hM3Dq-mCherry. Activation of the hM3Dq with CNO significantly boosted the activity of BCs, a subtype of MLIs that provide major inhibition on PCs (Fig. 1c, d). Due to synaptic inhibition, CNO decreased the firing rate and increased the firing variation of PCs that were connected to hM3Dq-expressing BCs. In contrast, CNO did not affect PCs that were innervated by non-transduced BCs (Fig. 1e, f). We also labeled PCs with an anti-Calbindin antibody to examine the BC-PC microcircuits and found that a BC typically innervated 3-5 neighboring PC somas (Supplementary Fig. 1c). These results are consistent with the original report on generation of the c-kit^{IRE5-Cre} mouse line ⁴.

It is not feasible to show how manipulating MLIs impacts the neuronal activity in the cerebellar nuclei (CN) using *ex vivo* electrophysiology. To maintain the spatial integrity of the MLI-PC circuits, sagittal sections are required for electrophysiological recordings. However, the PC-CN connections are aligned in a dorsal-ventral axis, perpendicular to the sagittal plane. Thus, the MLI-PC-CN pathway cannot be preserved in a single (sagittal or coronal) cerebellar section. As it is well established that PCs send GABAergic axons to CN neurons, we may infer that reduced PC activity releases inhibition on CN neurons and leads to increased tonic activity in the CN.

As to the transduction efficiency, we have addressed it in our previous responses. By normalizing the area containing mCherry to the total area of lobule IV/V or VI/VII in the midsagittal sections, we found that 29.2±2.7% of lobule IV/V and 35.8±3.0% of lobule VI/VII (n=6 mice/group) were affected (Results, Page 5). To explain how such a local manipulation would have an impact on behaviors, we have discussed that “Activation of MLIs in a lobule might not be sufficient to disturb general movement, even though it was sufficient to produce a SRM deficit. The sensitivity of the non-motor function to the cerebellar interference is congruent with other studies using similar approaches ^{7,8,27}” (Discussion, Page 14).

13) Figure 2d reports an increase in PC CV when light is ON. However, this is not so clear from the raw trace that is provided. Also, only a limited explanation is provided for how a change in CV is related to the overall phenotype(s) that is reported. Essentially, what does the increased CV mean?

Author Response: We thank the reviewer for catching this error in the presentation. We have replaced the old traces with more representative ones in a longer time frame to show the overall increase in the coefficient variation (CV) of inter-AP intervals in PCs due to the photostimulation (Fig. 2b, 2e). Further, by plotting a histogram with a Gaussian fit of the inter-AP intervals, we revealed that the stimulation created two distinct peaks centering around 14.5 ms and 54.2 ms, in contrast to the unimodal pattern with a peak around 17.5 ms without stimulation (Fig. 2f). Based on the mean values of the Gaussian distribution, the 1st peak likely denotes spontaneous PC activity (higher frequency) whereas the 2nd peak may correlate to the stimulus pattern (lower frequency). This finding is summarized in the Results (Page 8).

To explain the meaning of the increased CV, we have discussed that APs elicited from PCs are driven by PC intrinsic excitability and regulated by MLI inhibition, both of which can alter PC firing rate and regularity (measured with the CV). Clearly, future analysis is required to conclude

whether global changes in speed or specific patterns of PC activity are crucial for processing the information for SRM (Discussion, Page 15).

14) The cfos staining in Figure 3 is not so convincing. Perhaps the authors need to show additional adjacent tissue sections and they might also consider showing a larger area as well. As it stands, it is very hard to appreciate any difference from control. Last, some higher power images could also help in this regard.

Author Response: Thank you for raising this important point. We have shown the entire brain sections containing the regions of interest (ROI) in Fig. 3. Additionally, we have optimized the contrast of the ROI images with background subtraction to improve the visual representation (i.e., subtracting the mean intensity of the background within an unlabeled region in the same image). Lastly, we have added higher magnification images taken from the key brain regions in the SRM network (Supplementary Fig. 4). These sections were co-labeled with anti-CaMKII and anti-GAD67 antibodies for identifying the subtypes of c-Fos positive cells. More explanations are provided in our response to the next comment.

15) Related to above, what is the identity of the different cfos labeled populations? A little more information about what specific cells increase cfos expression in specific relevant areas would be very helpful.

Author Response: To identify subtypes of the c-Fos positive cells, we co-labeled glutamatergic and GABAergic neurons with respective anti-CaMKII and anti-GAD67 antibodies in brain sections fixed 90 min after the social recognition test, as described in Fig. 3a. We displayed the immunostaining images from several key brain regions in the SRM network such as the mPFC (PL), ACC, hippocampus (CA1), and amygdala (BLA) (Supplementary Fig. 4). c-Fos-positive cells in these regions were primarily glutamatergic neurons (~60-80%), with small proportions of GABAergic neurons (~10-20%) and other cell types (~10-20%). This result suggests that while various cells are activated, excitatory neurons that usually represent the output from each region play a major role in mediating the cellular response to SRM (Results, Page 10).

We have further explained this result in the context of the cerebellar engagement in retrieval, but not encoding, of SRM (Fig. 2). Memory encoding is the act of storing external features and internal states in a subset of neurons (engram) across various brain regions. Memory retrieval requires concurrent reactivation of the engram cells. Here, we showed the SRM task primarily activated excitatory neurons in the local circuits that were responsible for sending information to other brain regions, providing a mechanism for the interregional coactivation of engram cells (Discussion, Page 17). The fact that interfering with the vermis decreased brain-wide functional connectivity (Fig. 4) also indicates a unique position of the cerebellum in reinstatement of memory traces by coactivating the engrams. Together, your feedback has prompted us to gain new insights into the role of the cerebellum in SRM.

16) For Figure 6, it's unclear if different animals are shown across the different panels, or which panel goes with which animal. I think it would be useful to show multiple samples, with each case presented in a manner that one could track the projections per animal injected. Also, the example shown in 6b is complicated because it looks like both the fastigial and interposed are labeled. It would be more compelling to see more discrete targeting achieved for each nucleus.

Author Response: Thank you for your valuable input. We agree that in the early submission, the virus-based tracing spread from the fastigial nucleus (FN) to the interposed nucleus. To separate them, we have refined the neuronal tracing experiments by precisely targeting the FN. As PCs in the vermis mainly project to the FN¹, the new results represent a more accurate structural map of the vermal output (Fig. 6, 7, Supplementary Fig. 5).

The cerebellar output has been systematically described. For example, a recent study using a viral tracing technique specifically mapped out the FN output in the mouse brain². Our study was not designed to replicate previous anatomical reports but rather provide a structural basis for the cerebellar involvement SRM. For this purpose, we have focused on the brain regions highly relevant to SRM²⁸⁻³². While we did not track the FN projections in the entire brain, we have presented the coronal sections in the rostral-to-caudal order and co-labeled them with DAPI (a marker for cell nuclei) to better identify neuronal clusters in each region (Supplementary Fig. 5). The sections were outlined with the mouse brain atlas. Additionally, we have shown higher-magnification confocal images of all 24 brain regions in the SRM network (Fig. 6, 7).

The images were examples that represented the average fluorescent intensity from 3 different animals. For each region, 2-4 serial sections were examined, meaning 144-288 sections for 24 regions in 3 mice in total (2-4 x 24 x 3 = 144-288). Given the immense datasets, we feel that it would not benefit readers to show individual sections. Yet, we have addressed the concern about individual variability. To quantify the abundance of FN projections, we normalized the intensity of tdTomato labeling to the background in each region and summarized the results for 24 regions from 3 individual mice (Supplementary Fig. 5i). The original data can be found in the Source Data file. Our analysis indicates that the average fluorescent intensity in each region is quite consistent across individuals. As an example, raw images of the prelimbic cortex from the three animals are shown below.

Review Figure: Example images of the prelimbic cortex (PL) from three individual subjects. The axons were labeled via infusion of AAV1-hSyn-Cre in the fastigial nucleus of Ai9 mice.

The mouse brain in stereotaxic coordinates, compact third edition (Academic Press; 3rd edition, 2008).

References

- 1 Schmahmann, J. D., Oblak, A. L. & Blatt, G. J. in *Handbook of the Cerebellum and Cerebellar Disorders* (eds Mario U. Manto *et al.*) 605-624 (Springer International Publishing, 2022).
- 2 Fujita, H., Kodama, T. & du Lac, S. Modular output circuits of the fastigial nucleus for diverse motor and nonmotor functions of the cerebellar vermis. *Elife* **9**, doi:10.7554/eLife.58613 (2020).
- 3 Chao, O. Y., Zhang, H., Pathak, S. S., Huston, J. P. & Yang, Y. M. Functional Convergence of Motor and Social Processes in Lobule IV/V of the Mouse Cerebellum. *Cerebellum*, doi:10.1007/s12311-021-01246-7 (2021).
- 4 Amat, S. B. *et al.* Using c-kit to genetically target cerebellar molecular layer interneurons in adult mice. *PLoS One* **12**, e0179347, doi:10.1371/journal.pone.0179347 (2017).
- 5 Zingg, B. *et al.* AAV-Mediated Anterograde Transsynaptic Tagging: Mapping Corticocollicular Input-Defined Neural Pathways for Defense Behaviors. *Neuron* **93**, 33-47, doi:10.1016/j.neuron.2016.11.045 (2017).
- 6 Tervo, D. G. *et al.* A Designer AAV Variant Permits Efficient Retrograde Access to Projection Neurons. *Neuron* **92**, 372-382, doi:10.1016/j.neuron.2016.09.021 (2016).
- 7 Jackman, S. L. *et al.* Cerebellar Purkinje cell activity modulates aggressive behavior. *Elife* **9**, doi:10.7554/eLife.53229 (2020).
- 8 Zeidler, Z., Hoffmann, K. & Krook-Magnuson, E. HippoBellum: Acute Cerebellar Modulation Alters Hippocampal Dynamics and Function. *J Neurosci* **40**, 6910-6926, doi:10.1523/JNEUROSCI.0763-20.2020 (2020).
- 9 Strick, P. L., Dum, R. P. & Fiez, J. A. Cerebellum and nonmotor function. *Annu Rev Neurosci* **32**, 413-434, doi:10.1146/annurev.neuro.31.060407.125606 (2009).
- 10 Pisano, T. J. *et al.* Homologous organization of cerebellar pathways to sensory, motor, and associative forebrain. *Cell Rep* **36**, 109721, doi:10.1016/j.celrep.2021.109721 (2021).
- 11 Schmahmann, J. D. & Sherman, J. C. The cerebellar cognitive affective syndrome. *Brain* **121** (Pt 4), 561-579, doi:10.1093/brain/121.4.561 (1998).
- 12 Fastenrath, M. *et al.* Human cerebellum and corticocerebellar connections involved in emotional memory enhancement. *Proc Natl Acad Sci U S A* **119**, e2204900119, doi:10.1073/pnas.2204900119 (2022).
- 13 Cook, A. A., Fields, E. & Watt, A. J. Losing the Beat: Contribution of Purkinje Cell Firing Dysfunction to Disease, and Its Reversal. *Neuroscience* **462**, 247-261, doi:10.1016/j.neuroscience.2020.06.008 (2021).
- 14 Tsai, P. T. *et al.* Autistic-like behaviour and cerebellar dysfunction in Purkinje cell Tsc1 mutant mice. *Nature* **488**, 647-651, doi:10.1038/nature11310 (2012).
- 15 Cupolillo, D. *et al.* Autistic-Like Traits and Cerebellar Dysfunction in Purkinje Cell PTEN Knock-Out Mice. *Neuropsychopharmacology* **41**, 1457-1466, doi:10.1038/npp.2015.339 (2016).
- 16 Chao, O. Y. *et al.* Targeting inhibitory cerebellar circuitry to alleviate behavioral deficits in a mouse model for studying idiopathic autism. *Neuropsychopharmacology* **45**, 1159-1170, doi:10.1038/s41386-020-0656-5 (2020).
- 17 Yang, Y. M. *et al.* Identification of a molecular locus for normalizing dysregulated GABA release from interneurons in the Fragile X brain. *Mol Psychiatry* **25**, 2017-2035, doi:10.1038/s41380-018-0240-0 (2020).
- 18 Wang, S. S., Kloth, A. D. & Badura, A. The cerebellum, sensitive periods, and autism. *Neuron* **83**, 518-532, doi:10.1016/j.neuron.2014.07.016 (2014).
- 19 Hampson, D. R. & Blatt, G. J. Autism spectrum disorders and neuropathology of the cerebellum. *Front Neurosci* **9**, 420, doi:10.3389/fnins.2015.00420 (2015).

- 20 Yip, J., Soghomonian, J. J. & Blatt, G. J. Increased GAD67 mRNA expression in cerebellar interneurons in autism: implications for Purkinje cell dysfunction. *J Neurosci Res* **86**, 525-530, doi:10.1002/jnr.21520 (2008).
- 21 Simmons, D. H., Titley, H. K., Hansel, C. & Mason, P. Behavioral Tests for Mouse Models of Autism: An Argument for the Inclusion of Cerebellum-Controlled Motor Behaviors. *Neuroscience* **462**, 303-319, doi:10.1016/j.neuroscience.2020.05.010 (2021).
- 22 Heath, R. G., Dempsey, C. W., Fontana, C. J. & Myers, W. A. Cerebellar stimulation: effects on septal region, hippocampus, and amygdala of cats and rats. *Biol Psychiatry* **13**, 501-529 (1978).
- 23 Dietrichs, E. & Haines, D. E. Interconnections between hypothalamus and cerebellum. *Anat Embryol (Berl)* **179**, 207-220, doi:10.1007/bf00326585 (1989).
- 24 Zhu, J. N., Yung, W. H., Kwok-Chong Chow, B., Chan, Y. S. & Wang, J. J. The cerebellar-hypothalamic circuits: potential pathways underlying cerebellar involvement in somatic-visceral integration. *Brain Res Rev* **52**, 93-106, doi:10.1016/j.brainresrev.2006.01.003 (2006).
- 25 Hermans, E. J. *et al.* How the amygdala affects emotional memory by altering brain network properties. *Neurobiol Learn Mem* **112**, 2-16, doi:10.1016/j.nlm.2014.02.005 (2014).
- 26 McEwen, B. S., Nasca, C. & Gray, J. D. Stress Effects on Neuronal Structure: Hippocampus, Amygdala, and Prefrontal Cortex. *Neuropsychopharmacology* **41**, 3-23, doi:10.1038/npp.2015.171 (2016).
- 27 Badura, A. *et al.* Normal cognitive and social development require posterior cerebellar activity. *Elife* **7**, doi:10.7554/eLife.36401 (2018).
- 28 Hitti, F. L. & Siegelbaum, S. A. The hippocampal CA2 region is essential for social memory. *Nature* **508**, 88-92, doi:10.1038/nature13028 (2014).
- 29 Okuyama, T. Social memory engram in the hippocampus. *Neurosci Res* **129**, 17-23, doi:10.1016/j.neures.2017.05.007 (2018).
- 30 Garrido Zinn, C. *et al.* Major neurotransmitter systems in dorsal hippocampus and basolateral amygdala control social recognition memory. *Proc Natl Acad Sci U S A* **113**, E4914-4919, doi:10.1073/pnas.1609883113 (2016).
- 31 Tanimizu, T. *et al.* Functional Connectivity of Multiple Brain Regions Required for the Consolidation of Social Recognition Memory. *J Neurosci* **37**, 4103-4116, doi:10.1523/JNEUROSCI.3451-16.2017 (2017).
- 32 Phillips, M. L., Robinson, H. A. & Pozzo-Miller, L. Ventral hippocampal projections to the medial prefrontal cortex regulate social memory. *Elife* **8**, doi:10.7554/eLife.44182 (2019).

REVIEWERS' COMMENTS

Reviewer #1 (Remarks to the Author):

The authors have answered all my questions. This is an important contribution to understand the involvement of cerebellar vermi in social interactions.

Reviewer #2 (Remarks to the Author):

All my previous concerns have been appropriately addressed.

This is a wonderful paper, congratulations!

Point-by-point response to reviewers' comments on manuscript NCOMMS-22-07490A

Overall response: We wish to thank the reviewers for their valuable feedback on our manuscript, and for their enthusiasm about our study. Reviewers' comments are italicized, and our response to each point is outlined as follows.

REVIEWERS' COMMENTS

Reviewer #1 (Remarks to the Author):

The authors have answered all my questions. This is an important contribution to understand the involvement of cerebellar vermis in social interactions.

Author Response: We sincerely appreciate your enthusiasm about our study. Your insightful comments have greatly helped us to improve the quality of our manuscript.

Reviewer #2 (Remarks to the Author):

All my previous concerns have been appropriately addressed. This is a wonderful paper, congratulations!

Author Response: We sincerely appreciate your enthusiasm about our study. Your insightful comments have greatly helped us to improve the quality of our manuscript.